# EMBODIEDMAE: A UNIFIED 3D MULTI-MODAL REPRESENTATION FOR ROBOT MANIPULATION

## ABSTRACT

We present EmbodiedMAE, a unified 3D multi-modal representation for robot manipulation. Current approaches suffer from significant domain gaps between training datasets and robot manipulation tasks, while also lacking model architectures that can effectively incorporate 3D information. To overcome these limitations, we enhance the DROID dataset with high-quality depth maps and point clouds, constructing DROID-3D as a valuable supplement for 3D embodied vision research. Then we develop EmbodiedMAE, a multi-modal masked autoencoder that simultaneously learns representations across RGB, depth, and point cloud modalities through stochastic masking and cross-modal fusion. Trained on DROID-3D, EmbodiedMAE consistently outperforms state-of-the-art vision foundation models (VFMs) in both training efficiency and final performance across 70 simulation tasks and 20 real-world robot manipulation tasks on two robot platforms. The model exhibits strong scaling behavior with size and promotes effective policy learning from 3D inputs. Experimental results establish EmbodiedMAE as a reliable unified 3D multi-modal VFM for embodied AI systems, particularly in precise tabletop manipulation settings where spatial perception is critical.

## 1 INTRODUCTION

Pre-trained Vision Foundation Models (VFMs) have made remarkable progress in visual understanding (Caron et al., 2021; Oquab et al., 2024; He et al., 2022; Zhai et al., 2023; Nair et al., 2022; Majumdar et al., 2023; Bachmann et al., 2022; Zhu et al., 2025), becoming essential components for embodied AI systems (Octo Model Team et al., 2024; Kim et al., 2024; Black et al., 2024; Liu et al., 2025; Ze et al., 2024; Chi et al., 2023; Li et al., 2025b). As research demonstrates that 3D spatial understanding can improve robot manipulation capabilities (Ze et al., 2024; Ke et al., 2024; Li et al., 2025a; Zhen et al., 2024), the demand for effective 3D VFMs has grown. 3D information provides critical spatial context, enabling robots to localize targets, avoid collisions, and execute complex manipulations. Despite this increasing need, existing models fall short of meeting requirements.

There are two primary reasons behind the lack of suitable 3D VFMs for embodied AI. *First, a significant domain gap exists in training data*. Mainstream 3D VFMs are trained on outdoor or indoor static scenario datasets (Huang et al., 2023; Zhu et al., 2023; Qian et al., 2022; Yang et al., 2024a;b). These models operate at spatial scales incompatible with tabletop manipulation, resulting in a weak understanding of robot-object interactions (Ze et al., 2024). While training 3D embodied-specific VFMs from scratch on robot manipulation datasets seems promising, these efforts are hampered by extremely limited training data (Zhu et al., 2025; Qu et al., 2025; Vuong et al., 2023). *Second, there is a lack of efficient and scalable model architectures for 3D perception*. Simply integrating 3D information without careful design often degrades robot operation capabilities rather than enhancing. For example, many advanced 3D VFM architectures demonstrate unexpectedly poor performance in policy learning, sometimes even underperforming simple MLPs (Ze et al., 2024; Zhu et al., 2024).

To address these challenges, we propose EmbodiedMAE, a unified 3D multi-modal representation learning framework specifically designed for embodied AI. We first enhance the original DROID dataset (Khazatsky et al., 2024) by extracting high-quality metric depth maps and point clouds for each frame using ZED SDK temporal fusion and AI-augmented enhancement. This creates DROID-3D, a large-scale 3D robot manipulation dataset containing 76K trajectories (350 hours) of high-fidelity interaction data. This dataset provides the scale and quality needed for effective pre-training while maintaining domain compatibility with manipulation tasks. We then develop a

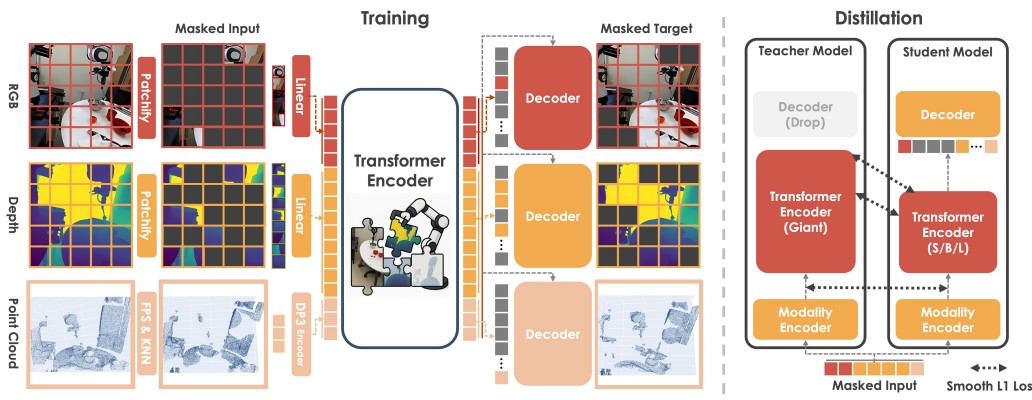

Figure 1: **Overview of EmbodiedMAE Pre-training.** We pre-train a ViT-Giant scale multi-modal MAE on the large-scale DROID-3D robot manipulation dataset. We fix the total number of unmasked patches across RGB, depth, and point cloud modalities. The mask ratio allocated to each modality is stochastically sampled. After the Giant model pre-training, we distill it to obtain our Small/Base/Large scale models.

multi-modal masked autoencoder that simultaneously learns representations across RGB images, depth maps, and point clouds through stochastic masking and cross-modal fusion. By masking different proportions of each modality and using explicit modal fusion in the decoder, our model learns to infer across modalities, developing powerful spatial perception capabilities and object-level semantic understanding (Figure 3).

To thoroughly validate our representation model, we conduct extensive evaluations across diverse settings: 40 tasks from the LIBERO benchmark (Liu et al., 2023) and 30 from the MetaWorld benchmark (Yu et al., 2019), 10 real-world tasks on the low-cost open-source SO100 robot (Cadene et al., 2024), and 10 on the high-performance xArm robot. We use a scaled-down RDT (Liu et al., 2025) model as the policy backbone to simulate the performance of VFMs in advanced VLA training, and compare EmbodiedMAE against various categories of state-of-the-art (SOTA) VFMs, including vision-centric models, language-augmented models, embodied-specific models, and 3D-aware models. Our experiments demonstrate that EmbodiedMAE consistently outperforms all baseline VFMs in both training efficiency and final performance, exhibits strong scaling behavior with model size, and effectively promotes policy from 3D input. These findings establish EmbodiedMAE as a reliable foundation model for embodied AI applications requiring robust 3D visual understanding.

Our contributions can be summarized as follows:

- We present EmbodiedMAE, a unified 3D multi-modal representation learning framework for embodied AI that effectively integrates RGB, depth, and point cloud modalities. It achieves SOTA performance in both RGB-only and multi-modal settings while maintaining computational efficiency and scaling properties.

- We introduce DROID-3D, a high-quality, large-scale DROID supplement containing 76K trajectories (350 hours) of robot data with synchronized RGB, depth maps, and point clouds. Unlike previous works that process subsets or use low-quality estimated depth, we provide temporally consistent depth by ZED SDK, creating a valuable resource for 3D robot learning.

- We establish comprehensive evaluation benchmarks for embodied representation learning across diverse settings: simulation tasks from LIBERO and MetaWorld, real-world tasks on a low-cost open-source robot (SO100), and tasks on a high-performance robot (xArm). Our results demonstrate consistent performance improvements across these varied platforms, validating the model's generalization capabilities.

## 2 METHODOLOGY

### 2.1 3D DATA COLLECTION

Effective pre-training of our model necessitates a large-scale 3D robot manipulation dataset. We conduct a systematic evaluation of depth data quality across several mainstream large-scale embodied AI datasets, primarily including BridgeDataV2 (Walke et al., 2023), RH20T (Fang et al., 2023), and

Figure 2: **Depth Quality Comparison.** We evaluate depth data quality across several mainstream large-scale embodied AI datasets. Both BridgeDataV2 and RH20T exhibit unreliable and noisy depth information. While prior work has explored the use of AI models for depth estimation, we observe that such methods lack temporal consistency. In contrast, our solution, ZED SDK processing, achieves superior and consistent depth quality.

DROID (Khazatsky et al., 2024), as illustrated in Figure 2. We find significant limitations in existing datasets: BridgeDataV2 contains only 13% data with 3D information, with available depth maps being of insufficient quality; RH20T exhibits similar issues with unreliable and noisy depth data; while DROID includes stereo image recordings but lacks readily usable 3D annotations. Several previous approaches attempted to address this by estimating depth from 2D images using AI models. For instance, SPA (Zhu et al., 2025) employs CrocoV2-Stereo (Weinzaepfel et al., 2023) to estimate depth for approximately 1/15 of the DROID dataset. We observe that such methods lack precision and temporal consistency, making them unable to accurately capture fine-grained details during robot-object interactions, which are essential for manipulation tasks.

To overcome these challenges, we use ZED SDK to extract the recording files in the raw DROID dataset. The ZED SDK integrates multiple techniques that significantly improve depth quality, including temporal fusion to reduce noise and increase consistency, AI-augmented enhancement to refine stereo matching in textureless regions, and hardware-calibrated metric depth to provide accurate absolute distance measurements. With these high-quality depth maps, we further extract point clouds with the camera's intrinsic matrix. We apply farthest point sampling (FPS) to downsample them to 8,192 points, striking a balance between computational efficiency and geometric fidelity. Unlike SPA's approach of processing only a subset of the DROID dataset, we process the complete collection of 76K trajectories (350 hours of interaction data), requiring nearly 500 hours of processing time. Due to these significant improvements in data quality and coverage, we construct DROID-3D as a supplementary resource to the original DROID dataset. We believe it will serve as a valuable resource for pre-training 3D VLA models and foster innovative research in embodied AI, particularly for applications requiring precise spatial understanding for manipulation tasks.

## 2.2 MULTI-MODAL ENCODER

EmbodiedMAE processes three modalities commonly used in robot perception: RGB images, depth maps, and point clouds. Given the robot observation of RGB image $I \in \mathbb{R}^{3 \times H \times W}$, depth $D \in \mathbb{R}^{1 \times H \times W}$, and point cloud $P \in \mathbb{R}^{M \times 3}$, we first use modal-specific patchifiers to project them into patches $\bar{I}, \bar{D}, \bar{P} \in \mathbb{R}^{L \times C}$. Then we draw a random binary mask for each modality $m_I, m_D, m_P \in \{0,1\}^L$, and obtain two complementary masked views $I_1 = \bar{I}[m_I], I_2 = \bar{I}[1 - m_I]$, similar for $D$ and $P$. We use a Vision Transformer (ViT) $f$ to process the unmasked patches and obtain the joint representation $h = f(I_1, D_1, P_1)$.

**Masking Strategies.** Effective masked autoencoding requires masking a large portion of input tokens during training, and the specific masking strategy has a significant impact on learned representations (Bachmann et al., 2022; He et al., 2022). Following Bachmann et al. (2022), we fix the total number of unmasked patches across all modalities, i.e., the number of ones in $(m_I, m_D, m_P)$ is fixed, and allocate them according to a symmetric Dirichlet distribution: $(\lambda_I, \lambda_D, \lambda_P) \sim \text{Dir}(\alpha)$, where $\lambda_I + \lambda_D + \lambda_P = 1$ and each $\lambda \geq 0$. The concentration parameter $\alpha$ controls the diversity of masking proportions. When $\alpha = 1$, the distribution is uniform over the simplex, assigning equal likelihood to all valid combinations. Lower values ($\alpha \ll 1$) tend to concentrate sampling on a single modality, while higher values ($\alpha \gg 1$) produce more balanced allocations across modalities. We intentionally avoid introducing any modality bias by keeping the distribution symmetric, aiming to maintain flexibility for a variety of downstream tasks and input configurations.

Figure 3: **EmbodiedMAE Visual Predictions.** We evaluate its visual predictions under three settings: **(a)** Two modalities are almost masked, leaving one modality as the major infer source (column 1-9). **(b)** Model predicts one modality from another one (column 10-11). **(c)** Model is allowed to see a modified RGB patch during depth-to-RGB prediction, where the color of the visible patch is altered (column 12).

**Modal Patchifiers.** For RGB and depth maps, we break them into $16 \times 16$-size patches, i.e., $L = \frac{H \cdot W}{16^2}$, and we incorporate 2D sine-cosine positional embeddings after a linear projection (Dosovitskiy et al., 2021; Touvron et al., 2021). For point clouds, we apply Farthest Point Sampling (FPS) to select $N$ cluster centers, and then use K-Nearest Neighbors (KNN) to group each center with its $K$ nearest neighbors, forming $N$ point groups of $K + 1$ points each, i.e., $L = N$. Each group is normalized and encoded using a DP3 encoder (Ze et al., 2024) to generate token embeddings, while each group center is processed by an MLP to create positional embeddings (Pang et al., 2022). We omit explicit modality-type embeddings, as the bias term in each projection layer implicitly encodes modality-specific information. These tokens are masked, concatenated, and passed to the ViT encoder.

**Transformer Encoder.** We implement the same ViT structure as DINOv2 (Oquab et al., 2024), with the exception of removing the [CLS] token. This design choice allows us to initialize the ViT directly from DINOv2 pre-trained weights, thereby enhancing its general capabilities.

## 2.3 MULTI-MODAL DECODER

The decoder is only used during EmbodiedMAE training, where it reconstructs the masked portions of each modality based on the visible tokens and learned [MASK] tokens.

Specifically, the decoder employs cross-attention to enable explicit fusion across modalities. Visible tokens from each modality are projected, concatenated with [MASK] tokens, and then augmented with positional embeddings to form the query sequence. Meanwhile, all visible patches are projected and enhanced with modality encodings to form the key and value sequences. The fused features are then fed into a smaller, modality-shared ViT decoder to produce the final hidden states. Modality-specific MLP heads generate the reconstruction outputs: masked RGB and depth patches, and normalized point coordinates for point cloud groups. Suppose that $(h_I, h_D, h_P) = f(I_1, D_1, P_1)$ are modality representations, the decoder outputs can be expressed as $g_I(h_I, h)$, $g_D(h_D, h)$, and $g_P(h_P, h)$, corresponding to each modality. Notably, our design shares transformer components across modalities, reducing computational cost by approximately a factor of three. We adopt a simple mean square error (MSE) loss:

$$\mathcal{L}_{\text{MAE}} = \mathbb{E}_{(I,D,P)\sim\mathcal{D},\text{Dir}(\alpha)} \left[ \underbrace{\|g(h_I, h) - I_2\|^2}_{\text{RGB}} + \underbrace{\|g(h_D, h) - D_2\|^2}_{\text{Depth}} + \underbrace{\|g(h_P, h) - P_2\|^2}_{\text{PointCloud}} \right], \quad (1)$$

where the decoder outputs $g_I(h_I, h)$, $g_D(h_D, h)$ are $l_2$-normalized, and $g_P(h_P, h)$ is group center-normalized, following that normalized targets yield better performance (He et al., 2022).

## 2.4 MODEL DISTILLATION

Following Oquab et al. (2024), we first train a ViT-Giant EmbodiedMAE model from scratch on the DROID-3D dataset, then distill it into Small, Base, and Large variants. Both teacher and student models receive identical masked inputs $(I_1, D_1, P_1)$, with the teacher model kept entirely frozen. Rather than simply copy the final outputs, we apply feature-level supervision at strategically selected

network depths to ensure comprehensive knowledge transfer. Specifically, we align features at three critical positions in the network hierarchy: (Bottom) immediately after the modal patchifiers to capture low-level perceptual features, (Top) at the final hidden layer to preserve high-level semantic understanding, and (Middle) at a middle layer positioned at 3/4 of the encoder depth to transfer intermediate representations (Bai et al., 2023) (For example, when distilling from a 24-layer ViT-L teacher to a 12-layer ViT-B student, the 9th layer of the student aligns with the 18th layer of the teacher.). We adopt trainable linear projections before computing alignment losses to accommodate dimensional differences between teacher and student features. Formally, we denote the feature alignment pairs $(y^j, h^j) \in A$, where $y^j$ and $h^j$ represent the $j$-th pair of hidden states from teacher and student models, respectively, and $l^j$ is the linear projector. The feature alignment loss is:

$$\mathcal{L}_{\text{Align}} = \sum\nolimits_{(y^j, h^j) \in A} \text{SmoothL1}\left(y^j, l^j(h^j)\right). \tag{2}$$

We train student models by jointly optimizing the standard multi-modal MAE reconstruction loss and the feature alignment loss (Figure 1, Distillation part):

$$\mathcal{L}_{\text{Distill}} = \mathcal{L}_{\text{MAE}} + \beta \cdot \mathcal{L}_{\text{Align}}, \tag{3}$$

where $\beta > 0$ controls the balance between mask autoencoding and feature alignment. This approach enables our smaller models to achieve performance closer to the Giant model while maintaining computational efficiency, making them practical in resource-constrained robotics applications.

## 2.5 Put All Together

Building on our architectural design described above, we first pre-train the Giant-scale model and subsequently distill it into more computationally efficient Small, Base, and Large variants on the DROID-3D dataset. We employ AdamW optimizer with a weight decay of 0.01. The base learning rate is set at 1.5e-4, incorporating an initial warmup period followed by a cosine schedule decay. We apply a 0.1 gradient norm clip to stabilize training. All computational workflows utilize `bfloat16` precision, which substantially reduces memory requirements and computational costs while maintaining numerical stability.

```python
from embodied_mae import EmbodiedMAEModel

model = EmbodiedMAEModel.from_pretrained("/path/to/ckpt")

rgb_feature  = model(rgb,  None,  None).last_hidden_states
# (b, 196, dim)
rgbd_feature = model(rgb, depth, None).last_hidden_states
# (b, 392, dim)
pc_feature   = model(None,  None,  pc).last_hidden_states
# (b, 196, dim)
```

Figure 4: **Usage Example.** We follow the Huggingface Transformers convention to make EmbodiedMAE highly user-friendly and easy to integrate.

During the pre-training phase, we maintain 96 unmasked patches across all modalities, representing approximately 1/6 of the total patch count. For the distillation phase, we further reduce the number of unmasked patches to 60, approximately 1/10 of the total. This extremely aggressive masking approach significantly decreases training costs without compromising representational quality, as the student models benefit from the teacher's already robust understanding of multi-modal relationships.

Our codebase follows Huggingface Transformers (Wolf et al., 2020) convention, making Embodied-MAE highly user-friendly. It ensures that researchers can easily incorporate our models into existing robotics pipelines with minimal adaptation effort. A simple usage example is illustrated in Figure 4.

## 3 Experiments

In this section, we present evaluation results of EmbodiedMAE across both simulation and real-world robotic manipulation tasks. Our experiments are designed to address three key research questions:

**(RQ1)** Does EmbodiedMAE learn features that integrate information across different modalities?

**(RQ2)** How does EmbodiedMAE perform compared to SOTA VFMs in robot manipulation tasks?

**(RQ3)** Can EmbodiedMAE enable efficient robot learning in real-world environments for both low-cost and high-performance robot platforms?

## 3.1 Experimental Setup

**Policy Network.** To evaluate how effectively different VFMs support advanced VLA models, we adopt a compact RDT (Liu et al., 2025) (approximately 40M parameters) as our policy network. This architecture has demonstrated excellent scalability and strong performance in diffusion-based

Table 1: **Success rate on MetaWorld benchmark.** We report the average success rate for each difficulty level. The numerical suffix following each level indicates the number of tested tasks. Note that **Average** row represents the average across all tasks rather than the three difficulty levels. Highest scores are emphasized with bold.

| MetaWorld Difficulty Level | R3M -RGB | SigLIP -RGB | DINOv2 -RGB | SPA -RGB | EmbodiedMAE -RGB | DINOv2 -RGBD | EmbodiedMAE -RGBD | DP3 -PointCloud | EmbodiedMAE -PointCloud |
|---|---|---|---|---|---|---|---|---|---|
| Easy (18) | 74.1 | 76.4 | 79.8 | 80.9 | 81.8 | 61.9 | **85.2** | 79.2 | 79.8 |
| Medium (9) | 28.1 | 32.7 | 57.1 | 62.8 | 60.4 | 35.6 | 63.2 | 48.0 | **76.7** |
| Very Hard (3) | 49.8 | 14.0 | 56.4 | 55.8 | 57.8 | **65.6** | 61.6 | 38.7 | 68.7 |
| **Average** | 57.9 | 57.0 | 70.7 | 73.0 | 73.0 | 54.4 | 76.2 | 65.8 | **77.7** |

policy learning. As shown in Figure 5, all baselines and EmbodiedMAE share the same architecture, ensuring fair comparison by isolating the visual representation component. See Section A.1 for more details.

**Baselines.** For a comprehensive comparison, we benchmark against several SOTA VFMs with diverse design principles: DINOv2-Large (Oquab et al., 2024) (vision-centric), SigLIP-Large (Zhai et al., 2023) (language-contrastive), R3M-Resnet50 (Nair et al., 2022), VC-1 (Majumdar et al., 2023), and SPA (Zhu et al., 2025) (embodied-specific). SPA incorporates implicit 3D priors during training, making it particularly relevant for comparison with our multi-modal approach.

**Benchmarks.** Our simulation evaluations are based on the LIBERO and MetaWorld benchmarks. LIBERO includes 40 tasks in four task suites: *Goal*, *Spatial*, *Object*, and *Long*. MetaWorld includes 30 tasks from various difficulty levels. For real-world experiments, we deploy the models

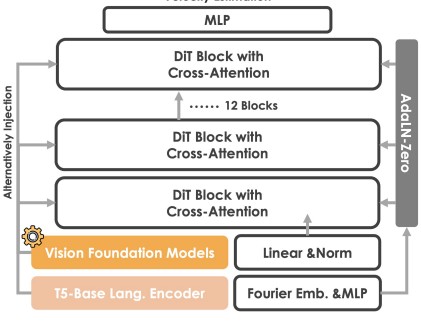

Figure 5: **Policy Network for All VFMs.** We adopt a compact RDT as the policy network, in which only VFMs are modular.

on two robot platforms: The SO100 robot (low-cost, open-sourced, equipped with dual RGB cameras) evaluated on 10 tasks in suites: *Pick&Place*, *MoveTo*, *Wipe*, and *Unfold*; The xArm robot (higher-precision, equipped with one Intel RealSense L515 LiDAR camera) evaluated on 10 tasks in suites: *Pick&Place*, *Pot*, *Pour*, and *Moka*. We show detailed task configurations in Section A.2.

### 3.2 MAE PREDICTIONS (RQ1)

To assess the ability of EmbodiedMAE to integrate information across modalities, we design a series of controlled experiments probing its cross-modal fusion capabilities. Our evaluation focuses on three settings: (a) Extreme modality inference: We mask most patches from two modalities, leaving primarily one modality as the inference source (Figure 3, columns 1-9). (b) Cross-modal translation: We test the model's ability to predict one entire modality from another, specifically RGB from depth (column 10) and depth from RGB (column 11). (c) Re-coloring: We allow the model to see a deliberately altered RGB patch during depth-to-RGB prediction (column 12), where the color of the visible patch is modified to assess semantic understanding. Our results demonstrate that EmbodiedMAE effectively leverages available modalities to reconstruct missing information, suggesting strong cross-modal alignment. In column 10, the predicted RGB from depth lacks precise color information but maintains structural fidelity, indicating the model has learned to separate geometric and appearance features. Similarly, in column 11, depth predictions from RGB show smoothed object boundaries compared to ground truth, revealing a learned prior for depth continuity. Most notably, in the re-coloring setting (column 12), when injecting an altered RGB patch during depth-to-RGB reconstruction, only the corresponding object (table) adopts the modified color while surrounding elements (background, robot, cup) maintain their original appearance. This suggests EmbodiedMAE has implicitly learned object-level semantic segmentation and can propagate semantic information based on contextual cues, despite never being explicitly trained for segmentation.

These visualizations collectively demonstrate that EmbodiedMAE possesses strong multi-modal fusion capabilities, enabling it to enhance spatial understanding in 3D embodied perception tasks.

### 3.3 OVERALL COMPARISON (RQ2)

In this section, we evaluate SOTA VFM baselines, EmbodiedMAE, and several its variants (in terms of model scale and input modality) on the LIBERO and MetaWorld benchmark. We report learning

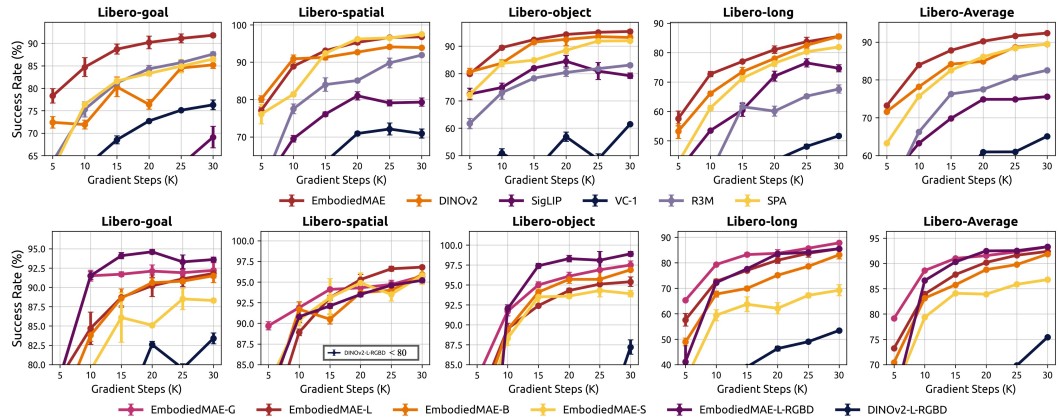

Figure 6: **Learning curve on LIBERO benchmark.** Each task is evaluated across 150 trials. Our model surpasses all baselines on the LIBERO benchmark and demonstrates scaling capabilities, with performance increasing proportionally with model size. Our model effectively leverages 3D information to further enhance policy performance, whereas naïvely incorporating depth information results in performance degradation.

curves on LIBERO in Figure 6 and success rate on MetaWorld in Section 3.3. *Unless otherwise specified, "EmbodiedMAE" refers to the Large-scale, RGB-only variant.*

*Finding 1:* **EmbodiedMAE consistently outperforms all baseline VFMs in terms of both training efficiency and final performance.** Among the baselines, SPA and DINOv2 are the most competitive ones. SPA shows score gains on tasks where spatial understanding is crucial, e.g., LIBERO-Spatial and MetaWolrd, and performs comparably to DINOv2. The language-contrastive model, SigLIP, performs poorly across all embodied tasks, consistent with findings from Zhu et al. (2025). R3M and VC-1, although specifically designed for robot learning, do not demonstrate clear advantages.

*Finding 2:* **EmbodiedMAE exhibits strong scaling behavior with model size.** Performance improves monotonically as model capacity increases. Among all the variants, only the Small variant shows unstable performance on LIBERO-Goal and LIBERO-Object suites. The Base and Large models achieve similar performances, with the Large model slightly ahead. The Giant model consistently delivers superior performance, particularly in training efficiency. These results suggest EmbodiedMAE to be an effective training paradigm for scaling multi-modal representation learning.

*Finding 3:* **EmbodiedMAE promotes policy learning from 3D input.** When provided with RGBD inputs, EmbodiedMAE establishes a substantial performance gap over other baselines on both LIBERO and MetaWorld benchmarks. Remarkably, our Large-scale RGBD model even outperforms the Giant-scale RGB-only model on LIBERO-Goal and LIBERO-Object suites, and performs comparably on average across the LIBERO benchmark. In contrast, adding a trainable depth branch for DINOv2 (See Section A.3 for details of this variant) can degrade performance relative to RGB-only input, consistent with observations in Zhu et al. (2024). These findings establish EmbodiedMAE as a reliable VFM for scenarios requiring 3D visual understanding.

### 3.4 REAL-WORLD EXPERIMENTS (RQ3)

To further assess generalization in practical settings, we conduct real-world evaluations on two robot platforms: the low-cost, open-source SO100 (Cadene et al., 2024) and the high-performance xArm. We show quantitative results in Figure 8, and rollout visualizations in Figure 7.

*Finding 1:* **EmbodiedMAE maintains SOTA performance in real-world robot manipulation.** EmbodiedMAE consistently achieves SOTA performance across real-world manipulation tasks, particularly those requiring strong spatial understanding. With multi-modal inputs, EmbodiedMAE further improves policy learning performance: EmbodiedMAE-RGBD and EmbodiedMAE-PC both surpass naïve fusion baselines such as DINOv2-RGBD (Section A.3) and DP3 (Ze et al., 2024), highlighting the effectiveness of our design in promoting robust 3D perception for real-world robotics.

*Finding 2:* **3D information plays a critical role in robot manipulation.** Incorporating 3D inputs significantly improves task success rates. We observe that most failures in baseline models stem from localization errors, causing grasp failures or collisions. EmbodiedMAE-RGBD, benefiting from

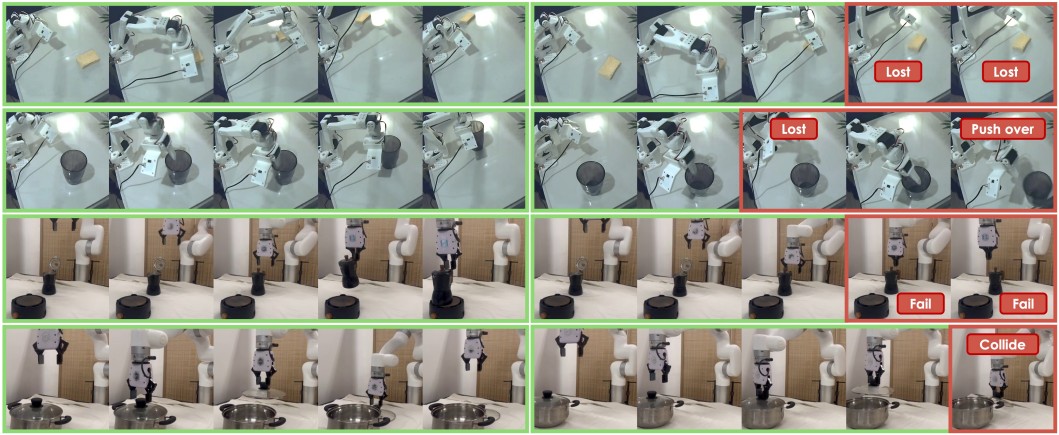

Figure 7: **Successful rollouts of EmbodiedMAE (Left) and typical failure cases of baselines (Right).** Baseline models often fail due to inaccurate localization, leading to object loss, grasp failure, or collisions. In contrast, EmbodiedMAE benefits from stronger spatial perception and avoids such errors more effectively.

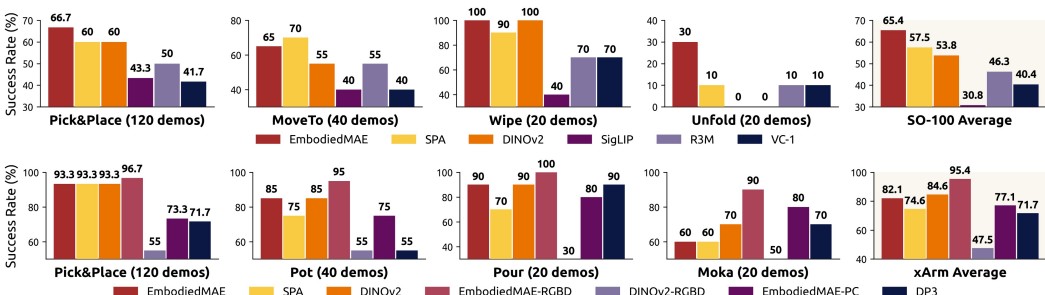

Figure 8: **Evaluation results on SO100 and xArm platforms.** Each task is evaluated across 10 trials. On the SO100 platform, our model outperformed all baselines in the RGB-only setting. On the xArm platform, our model achieved comparable performance to SOTA baselines in the RGB-only setting, while significantly surpassing baselines in both RGBD and Point Cloud settings.

enhanced spatial understanding, avoids these issues more reliably (see Figure 7). The choice of 3D modality also matters. Although prior works (Li et al., 2025a; Ze et al., 2024; Zhu et al., 2024) have highlighted the compactness and training efficiency of point cloud (PC) representations, we find their practical effectiveness is hindered by sensor noise from object reflectivity and lighting variations. Consequently, PC-based policies even underperform RGB-only inputs. In contrast, the RGBD setting, where depth serves as an auxiliary cue, yields better performance and is more robust to noise. This suggests that effective post-processing of PCs is essential for leveraging them reliably.

### 3.5 ABLATION STUDIES

Due to the prohibitive cost of ViT-Giant pre-training, our ablation studies focus on model distillation insights. We evaluate masking ratio, feature alignment, and loss ratio on the LIBERO benchmark, reporting average success rates in Table 4, with default settings underlined. **(1) Masking Ratio**: Our default configuration sets 60 unmasked patches, approximately masking ratio of 90%. We test 70%, 80%, and 100% ratios (100% representing training with only feature alignment loss). Results indicate performance insensitivity to masking ratio, though ratios ¡100% perform better, suggesting feature alignment's predominant role while mask autoencoding provides additional benefits. **(2) Feature Alignment**: By default, we implement feature alignment at three positions (see Section 2.4). Sequential removal of alignment points reveals diminishing impact from Top to Bottom, with each component contributing positively to model performance. **(3) Loss Ratio**: With default $\beta = 1$, we test $\beta = 0.5/2.0/4.0$. Results show performance robustness across $\beta$ values, with slight degradation at $\beta < 1.0$, confirming feature alignment necessity, consistent with findings in (Bai et al., 2023). **(4) Policy Model**: While our primary policy focus is on diffusion-based models, we recognize the popularity of transformer-based models like the Action Chunking Transformer (ACT) (Fu et al.,

2024). To confirm the generalizability of EmbodiedMAE's representations, we expand our evaluation to include the ACT on LIBERO-Goal (RGB and RGBD) and MetaWorld (PC) benchmarks.

Table 2: **Ablation study with ACT policy on LIBERO-Goal.**

| Policy Model | EmbodiedMAE | DINOv2 | SPA | EmbodiedMAE-RGBD | DINOv2-RGBD |
|---|---|---|---|---|---|
| ACT Policy | 83.7 | 76.3 | 82.5 | **90.8** | 82.2 |

Table 3: **Ablation study with ACT policy on MetaWorld.**

| Policy Model + VFM | Easy (18) | Medium (9) | Very Hard (3) |
|---|---|---|---|
| ACT Policy + EmbodiedMAE-PC | **80.0** | **64.4** | **56.2** |
| ACT Policy + DP3 | 78.8 | 42.7 | 33.1 |

## 4 RELATED WORKS

**Vision Foundation Models** are models trained on large-scale data in a self-supervised or semi-supervised manner that can be adapted for several other downstream tasks (Bommasani et al., 2022). Beyond conventional image classification, these models have shown strong transfer capabilities to tasks such as depth estimation (Yang et al., 2024a;b; Weinzaepfel et al., 2023), semantic segmentation, and robot control (Octo Model Team et al., 2024; Kim et al., 2024; Liu et al., 2025; Kim et al., 2025). Common pre-training techniques include contrastive learn-

| Masking Ratio | 0.7 | 0.8 | 0.9 | 1.0 |
|---|---|---|---|---|
|  | 92.2 | 91.2 | 92.4 | 90.1 |
| Feature Alignment | w/o Bottom | w/o Middle | w/o Top | All |
|  | 91.4 | 88.5 | 74.4 | 92.4 |
| Loss Ratio $\beta$ | 0.5 | 1 | 2 | 4 |
|  | 90.8 | 92.4 | 91.1 | 92.2 |

Table 4: **Ablation study on LIBERO.** We conduct ablation experiments on masking ratio, feature alignment, and loss ratio on the LIBERO benchmark and report the average success rate.

ing (He et al., 2019; Chen et al., 2020; Chen* et al., 2021), masked autoencoding (Bai et al., 2023; Tong et al., 2022; Wang et al., 2023; Feichtenhofer et al., 2022; He et al., 2022), self-distillation (Oquab et al., 2024; Caron et al., 2021), and CLIP-style language-image contrastive learning (Zhai et al., 2023; Radford et al., 2021). VFMs greatly improve AI systems' visual understanding.

**Visual Representations for Embodied AI** are crucial for enabling agents to perceive and interact with the physical world. Embodied perception must model robot-object interactions in dynamic environments, which general-purpose VFMs trained on static images often lack. Several recent methods have attempted to bridge this gap by training models directly on robot datasets. However, the limited scale and quality of embodied data hinder their generalization. These embodied-specific models often fail to generalize as well as VFMs trained on diverse in-the-wild datasets. As a result, many VLA models still rely on general-purpose VFMs like DINOv2 (Oquab et al., 2024; Kim et al., 2024; 2025) and SigLIP (Zhai et al., 2023; Liu et al., 2025; Kim et al., 2024) for better generalization, prompting the need for dedicated large-scale embodied VFM pretraining.

**3D Robot Learning** has proven effective in improving both embodied agents' training efficiency and policy performance (Ze et al., 2024; Li et al., 2025a; Zhu et al., 2024). Properly introducing 3D visual inputs often leads to better spatial understanding compared to RGB-only inputs. However, naïvely incorporating 3D information, e.g., adding an extra depth channel, may severely degenerate the model's performance. Scalable native 3D multi-modal models remain largely absent in the current research landscape. EmbodiedMAE aims to address this gap by pre-training VFMs on large-scale, embodied-specific datasets to facilitate the development of scalable and effective 3D VLA models.

## 5 CONCLUSION, LIMITATIONS, AND FUTURE WORKS

In this work, we introduce EmbodiedMAE, a unified 3D multi-modal representation learning framework designed for embodied AI. We first construct DROID-3D, a high-quality, large-scale DROID supplement. Then we propose a multi-modal masked autoencoder architecture that fuses RGB, depth, and point cloud inputs through stochastic masking and cross-modal decoding. Trained on DROID-3D, our model, EmbodiedMAE, demonstrates superior spatial understanding, strong multi-modal fusion ability, and effective scaling behavior. It outperforms strong VFM baselines across 70 simulation tasks and 20 real-world tasks on two robot platforms (SO100 and xArm). We believe both the DROID-3D dataset and EmbodiedMAE provide a valuable resource for 3D robot learning research. Despite the strong performance, EmbodiedMAE remains solely a vision backbone and does not natively support language instruction as input. A promising future direction is to fully leverage the language and action annotations available in the DROID-3D dataset to train a vision-language backbone, or even develop a multi-modal VLA model for instruction-following general embodied agents.

## 6 ETHICS STATEMENT

The authors have adhered to the ICLR Code of Ethics. This work does not involve human subjects, sensitive data, or raise any direct ethical concerns. All datasets used are publicly available.

## 7 REPRODICIBILITY STATEMENT

We are committed to ensuring the reproducibility of our research. Our source code will be made public upon publication. Detailed descriptions of our methods and model architectures are available in the section Section 2.5. All experimental settings, including datasets, training hyperparameters, and evaluation settings, are specified in the section Section 3 and Appendix.

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

# A DETAILS OF EXPERIMENTAL SETUP

## A.1 POLICY NETWORK

To evaluate how well Vision Foundation Models (VFMs) support advanced Vision-Language Action (VLA) models, we use the RDT (Liu et al., 2025) architecture as our evaluation policy network, which has demonstrated excellent scalability and strong performance in diffusion-based policy learning. Diffusion timestamps and robot kinematic information are integrated into the policy network using AdaLN-Zero (Peebles & Xie, 2022). The vision and language embeddings are used as the Keys and Values in the cross-attention layers to be integrated into the policy network alternately (Liu et al., 2025). The Transformer architecture has a hidden dimension of 384, with 6 attention heads, and 12 layers.

For action generation, we use a flow-matching model similar to (Black et al., 2024). Diffusion timestamps are treated as continuous values within the range $[0, 1]$; we do not discretize them. Instead, they are represented using a Fourier embedding with a scale of 0.2 (Dong et al., 2024). During training, diffusion timestamps are sampled from a uniform distribution over the interval $[0, 1]$. For inference, we solve the corresponding ODE using the Euler method, dividing the interval $[0, 1]$ into equal-sized steps.

## A.2 DETAILS OF BENCHMARKS

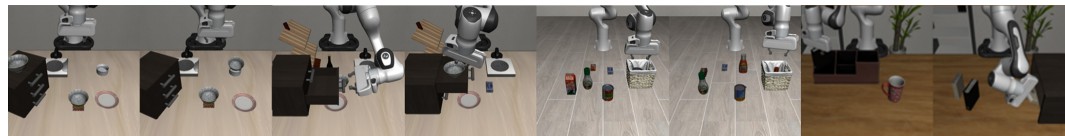

Figure 9: **LIBERO simulation benchmark.** We conduct experiments on 40 tasks from four task suites in the LIBERO benchmark. We show two task examples for each suite here.

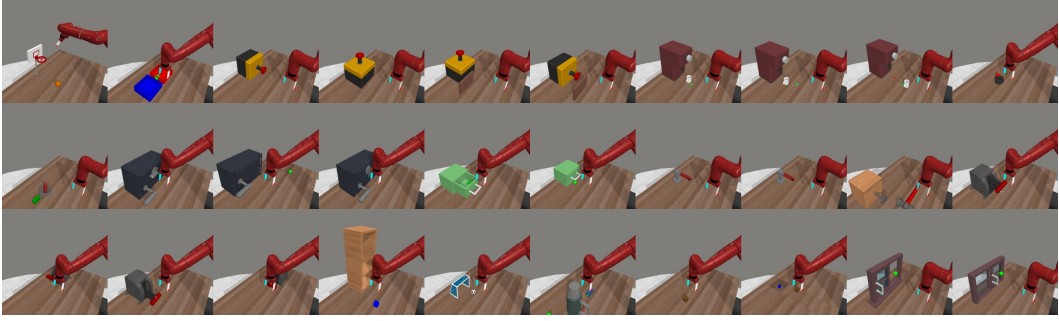

Figure 10: **MetaWorld simulation benchmark.** We conduct experiments on 30 tasks of three difficulty levels in the MetaWorld benchmark. We show all task examples here.

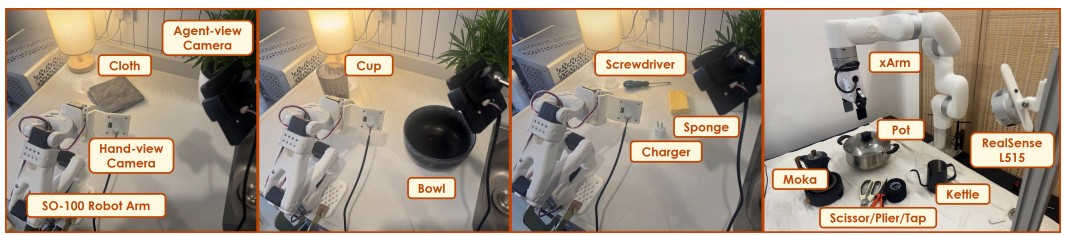

Figure 11: **Real-world experimental setups.** We conduct experiments on both SO100 and xArm platform. For each robot, we design a suite of 10 tabletop tasks involving diverse objects.

Table 5: Task description of each task in the LIBERO benchmark.

| Task Suite | Task Description |
|---|---|
| LIBERO-Goal | open the middle layer of the drawer
put the bowl on the stove
put the wine bottle on the top of the drawer
open the top layer of the drawer and put the bowl inside
put the bowl on the top of the drawer
push the plate to the front of the stove
put the cream cheese on the bowl
turn on the stove
put the bowl on the plate
put the wine bottle on the rack |
| LIBERO-Spatial | pick the akita black bowl between the plate and the ramekin and place it on the plate
pick the akita black bowl next to the ramekin and place it on the plate
pick the akita black bowl from table center and place it on the plate
pick the akita black bowl on the cookies box and place it on the plate
pick the akita black bowl in the top layer of the wooden cabinet and place it on the plate
pick the akita black bowl on the ramekin and place it on the plate
pick the akita black bowl next to the cookies box and place it on the plate
pick the akita black bowl on the stove and place it on the plate
pick the akita black bowl next to the plate and place it on the plate
pick the akita black bowl on the wooden cabinet and place it on the plate |
| LIBERO-Object | pick the alphabet soup and place it in the basket
pick the cream cheese and place it in the basket
pick the salad dressing and place it in the basket
pick the bbq sauce and place it in the basket
pick the ketchup and place it in the basket
pick the tomato sauce and place it in the basket
pick the butter and place it in the basket
pick the milk and place it in the basket
pick the chocolate pudding and place it in the basket
pick the orange juice and place it in the basket |
| LIBERO-Long | put both the alphabet soup and the tomato sauce in the basket
put both the cream cheese box and the butter in the basket
turn on the stove and put the moka pot on it
put the black bowl in the bottom drawer of the cabinet and close it
put the white mug on the left plate and put the yellow and white mug on the right plate
pick up the book and place it in the back compartment of the caddy
put the white mug on the plate and put the chocolate pudding to the right of the plate
put both the alphabet soup and the cream cheese box in the basket
put both moka pots on the stove
put the yellow and white mug in the microwave and close it |

Table 6: Task description of each task in the MetaWorld benchmark.

| Task Name | Task Description |
|---|---|
| basketball | Dunk the basketball into the basket. |
| bin-picking | Grasp the puck from one bin and place it into another bin. |
| button-press | Press a button. |
| button-press-topdown | Press a button from the top. |
| button-press-topdown-wall | Bypass a wall and press a button from the top. |
| button-press-wall | Bypass a wall and press a button. |
| coffee-button | Push a button on the coffee machine. |
| coffee-pull | Pull a mug from a coffee machine. |
| coffee-push | Push a mug under a coffee machine. |
| dial-turn | Rotate a dial 180 degrees. |
| disassemble | Pick a nut out of the peg. |
| door-lock | Lock the door by rotating the lock clockwise. |
| door-open | Open a door with a revolving joint. |
| door-unlock | Unlock the door by rotating the lock counter-clockwise. |
| drawer-close | Push and close a drawer. |
| drawer-open | Open a drawer. |
| faucet-close | Rotate the faucet clockwise. |
| faucet-open | Rotate the faucet counter-clockwise. |
| hammer | Hammer a screw on the wall. |
| handle-press | Press a handle down. |
| handle-press-side | Press a handle down sideways. |
| handle-pull | Pull a handle up. |
| handle-pull-side | Pull a handle up sideways. |
| shelf-place | Pick and place a puck onto a shelf. |
| soccer | Kick a soccer into the goal. |
| stick-push | Grasp a stick and push a box using the stick. |
| sweep | Sweep a puck off the table. |
| sweep-into | Sweep a puck into a hole. |
| window-close | Push and close a window. |
| window-open | Push and open a window. |

Table 7: **Task description of each task in the SO100 and xArm benchmark.** As each parameter combination introduces one task, each task suite has 10 tasks in total. For each task, we test the model for 10 trials.

| Task Suite | Task Description | Parameter |
|---|---|---|
| SO100 | pick [A] and place it on the [B] side of the table | [A]: ["screwdriver", "sponge", "charger"],[B]: ["left", "right"] |
| | move [A] to the center of the table | [A]: ["cup", "bowl"] |
| | pick the cloth and wipe the table | None |
| | unfold the cloth | None |
| xArm | pick [A] and place it on the [B] side of the table | [A]: ["scissor", "plier", "tap"],[B]: ["left", "right"] |
| | open the pot lid or put the lid on the pot | [open, close] |
| | pour the water from the kettle into the cup | None |
| | place the Moka pot on the cooker | None |

**LIBERO.** The LIBERO simulation benchmark (Liu et al., 2023) features a Franka Emika Panda arm in simulation across four challenging task suites: *Goal*, *Spatial*, *Object*, and *Long*. Each suite comprises 10 tasks with 500 demonstrations and is designed to investigate controlled knowledge transfer related to goal variations, spatial configurations, object types, and long-horizon tasks. Unlike prior work (Kim et al., 2024; 2025), we do not filter out unsuccessful demonstrations, aiming for a more realistic evaluation setting. For policy training, the model predicts action chunks of length 16; after each chunk prediction, 8 steps are executed before generating the next chunk. The observation space includes 2-view RGB images at the current time step, without historical observations. During evaluation, following Liu et al. (2023), each task is tested over 50 trials with 3 different random seeds, and success rates are reported. To provide a clearer understanding of the task suites, we present agent-view observations in Figure 9 and detailed task descriptions in Table 5.

**MetaWorld.** The MetaWorld simulation benchmark (Yu et al., 2019) includes 50 distinct tabletop manipulation tasks using a Sawyer robot arm. We select 30 tasks from *easy*, *medium*, and *very hard* difficulty levels to evaluate VLA models. We use a scripted policy to collect 20 demonstrations for each task. For policy training, the model predicts action chunks of length 16; after each chunk prediction, 16 steps are executed before generating the next chunk. The observation space consists of a single RGB image at the current time step, without historical observations. During evaluation, each task is tested over 50 trials with 3 different random seeds, and success rates are reported. To better illustrate the task suites, we show agent-view observations in Figure 10 and task descriptions in Table 6.

**SO100 Robot Manipulation.** The SO100 robot (Cadene et al., 2024) is a low-cost, open-source 6-DoF manipulator, with both the leader and follower arms costing approximately $250. We assemble the hardware using a 3D-printed kit provided by the open-source community. The robot has two RGB cameras: one mounted on the wrist and the other positioned to provide a third-person view. Both cameras operate at a resolution of 640×480 and 25 FPS. The robot controller runs at 30Hz, and actions are defined as target absolute joint angles. Due to its low-cost design, the platform has several hardware limitations, including significant arm jitter, low load capacity, and occasional camera lag, which present practical challenges for developing embodied AI systems. However, given the increasing adoption of such affordable open-source robots by the research community, we believe that evaluating models on these lower-performance systems offers valuable insights and broader applicability. We design four categories of tabletop manipulation tasks for the SO100 setup: *(1) Pick&Place:* involving 3 objects and 2 placement zones (6 tasks), *(2) MoveTo:* navigating 2 objects to a single target zone (2 tasks), *(3) Wipe:* picking up a cloth and wiping the table (1 task), and *(4) Unfold:* unfolding a cloth (1 task). In total, we evaluate performance on 10 distinct tasks. Language instructions for each task are listed in Table 7, and visual examples of the task environments are shown in Figure 11.

During data collection, we record 20 demonstrations per task. For policy training, the model predicts an action chunk of length 64; after each chunk prediction, 40 steps are executed before generating the next chunk. The observation space includes 2-view RGB images at the current time step, along with the absolute joint angles from the current and previous 10 steps. During evaluation, each task is tested over 10 trials, and success rates are reported.

**xArm Robot Manipulation.** xArm is a high-performance 7-DoF manipulator. The robot is equipped with a third-person view Intel RealSense L515 LiDAR camera, operating at 640×480 resolution and 30 FPS. We collect both RGB and depth images from the camera. The robot controller runs at 30Hz, and actions are defined as target absolute joint angles. We design four categories of tabletop manipulation tasks for the xArm setup: *(1) Pick&Place:* involving 3 objects and 2 placement zones (6 tasks), *(2) Pot:* taking off or putting on the pot lid (2 tasks), *(3) Pour:* pouring water from the kettle into the cup (1 task), and *(4) Moka:* placing the Moka pot on the cooker (1 task). In total, we evaluate performance on 10 distinct tasks. Language instructions for each task are listed in Table 7, and visual examples of task environments are shown in Figure 11.

During data collection, we record 20 demonstrations per task. For policy training, the model predicts an action chunk of length 64; after each chunk prediction, 40 steps are executed before generating the next chunk. The observation space includes a third-person view RGB image at the current time step, as well as the absolute joint angles from the current and previous 10 steps. During evaluation, each task is tested over 10 trials, and success rates are reported.

Table 8: **Hyperparameters for EmbodiedMAE training.** Since we use pre-training and distillation for different model scales, we use **(P)** to denote pre-training hyperparameters and **(D)** to denote distillation hyperparameters.

| Hyperparameters | Values |
|---|---|
| GPUs | 8xNVIDIA L40 (60GB) **(P)** or 4xNVIDIA Geforce RTX4090 (24GB) **(D)** |
| learning rate | 3e-4 peak LR (500 steps linear warmup, 300k steps cosine decay to 3e-6) |
| batch size | 512 |
| training steps | 200K **(P)** 100K **(D)** |
| input modalities | 224x224x3 RGB images, 224x224 Depth maps, 8,192 Point Clouds |
| image augmentations | ColorJitter(brightness=0.1, contrast=0.1, saturation=0.1, hue=0.05) |
| trainable parameters | 1.1B Giant **(P)** 304M Large, 87M Base, 22M Small **(D)** encoders, and 44M decoders |
| mask ratio | 84% **(P)** 90% **(D)** |
| Distillation $\beta$ | 1.0 |

### A.3 DINOv2-RGBD BASELINE

To establish a reliable and effective RGBD baseline for practical applications, we follow the approach outlined in Zhu et al. (2024), designing a method that naïvely incorporates depth information based on DINOv2. We introduce an additional Conv2D layer to patchify the depth map, summing the resulting patches with DINOv2's RGB patchifying output before encoding through the DINOv2 Encoder. We initialize the depth patchifier's weights and biases to zero, ensuring that the representation model remains functionally equivalent to DINOv2 at the beginning of training. During training, we update only the depth patchifier's gradients, allowing depth information to be learned following DINOv2's prior knowledge.

## B IN-DEPTH ANALYSIS OF POINT CLOUD MODALITY

In Section 3, we observed that while the Point Cloud (PC) modality performs robustly in simulation, it faces challenges in real-world deployment, primarily due to the domain gap introduced by sensor noise. To investigate the root causes of this discrepancy and validate our design choices, we conduct a series of supplementary experiments on the xArm platform and the MetaWorld benchmark. These experiments focus on three key aspects: comparison with other 3D PC Vision Foundation Models (VFMs), the impact of the encoder architecture, and the necessity of high-quality data pre-processing.

### B.1 COMPARISON WITH GENERAL 3D VFMS

To contextualize the performance of EmbodiedMAE, we investigate whether the performance drop in the real-world PC setting is specific to our model or a broader issue with current PC VFMs. We compare our approach against PonderV2 (Zhu et al., 2023), a state-of-the-art 3D VFM pre-trained on large-scale indoor and outdoor scene datasets. As shown in Table 9, PonderV2 achieves a success rate of 54.1% on the xArm benchmark, significantly underperforming both EmbodiedMAE (77.1%) and the baseline DP3 policy (71.7%). We attribute this to the substantial domain gap between the pre-training data of general 3D VFMs (large-scale scenes) and the downstream task (precise tabletop manipulation). This finding reinforces the necessity of pre-training on domain-specific robotic data, such as DROID-3D, to learn representations that are effective for manipulation tasks.

### B.2 DESIGN CHOICE OF POINT CLOUD ENCODER

In our default architecture, we employ a lightweight encoder consisting of Farthest Point Sampling (FPS), K-Nearest Neighbors (KNN), and a DP3-style MLP encoder (Ze et al., 2024). This design choice was motivated by the hypothesis that the primary role of the modality-specific encoder is tokenization, while complex feature extraction and cross-modal fusion should occur within the heavy Transformer backbone. To challenge this hypothesis, we replace our lightweight PC encoder with a computationally heavier sparse convolutional network (Contributors, 2022). We train an *EmbodiedMAE-PC (SparseConv)* variant by distilling from the EmbodiedMAE-Giant model (omitting the modality encoder distillation loss due to the architectural change).

The results, presented in Table 9 and Table 10, indicate no significant performance advantage for the SparseConv encoder. On the xArm benchmark, the performance marginally increases to 79.2%, and on MetaWorld, it remains comparable to the default encoder. This observation validates our design philosophy: feature extraction is predominantly handled by the Transformer backbone, effectively making the specific architecture of the tokenizer less critical to the final policy performance.

### B.3 IMPACT OF DATA QUALITY AND PRE-PROCESSING

Finally, we hypothesize that the sensitivity of the PC modality stems from the quality of the input data. Real-world depth sensors (e.g., Intel RealSense L515) often produce noisy depth maps with outliers, which differ from the high-fidelity, processed data seen during pre-training. To address this, we implement an enhanced pre-processing pipeline for the xArm platform. This pipeline includes: (1) applying radius outlier removal to the raw depth map; (2) utilizing metric depth estimated by CrocoV2-Stereo Weinzaepfel et al. (2023) to guide a filtering process on the raw depth; (3) converting the refined depth to a point cloud; and (4) applying voxel downsampling followed by FPS.

The results of this *Enhanced PC Quality* setting are compelling. As shown in Table 9, EmbodiedMAE-PC sees a substantial performance boost, reaching an average success rate of 82.1% (up from 77.1%). Conversely, the DP3 baseline, which is trained from scratch, shows negligible improvement (72.9% vs. 71.7%). This suggests that while scratch-trained policies can overfit to noisy distributions, pre-trained VFMs like EmbodiedMAE rely on high-quality structural information to align with their learned representations. Therefore, improving input quality is a highly effective strategy for unlocking the potential of 3D VFMs in real-world scenarios.

Table 9: Ablation studies on the xArm real-world platform. We compare EmbodiedMAE against PonderV2, evaluate the impact of replacing the lightweight encoder with SparseConv, and assess the benefits of enhanced point cloud quality.

| Method | Pick&Place | Pot | Pour | Moka | Average |
|---|---|---|---|---|---|
| *Standard Point Cloud Input* | | | | | |
| DP3 Encoder (Ze et al., 2024) | 71.7 | 55.0 | 90.0 | 70.0 | 71.7 |
| PonderV2 (Zhu et al., 2023) | 66.7 | 40.0 | 70.0 | 40.0 | 54.1 |
| **EmbodiedMAE-PC (Ours)** | 73.3 | 75.0 | 80.0 | 80.0 | **77.1** |
| EmbodiedMAE-PC (SparseConv) | 71.6 | 85.0 | 80.0 | 80.0 | 79.2 |
| *Enhanced Point Cloud Quality* | | | | | |
| DP3 (Enhanced) | 76.7 | 55.0 | 90.0 | 70.0 | 72.9 |
| **EmbodiedMAE-PC (Enhanced)** | **93.3** | **85.0** | 80.0 | 70.0 | **82.1** |

Table 10: Supplementary ablation on MetaWorld. Replacing the default PC encoder with SparseConv yields negligible performance gains, confirming that the Transformer backbone dominates feature extraction.

| Method | Easy (18) | Medium (9) | Very Hard (3) |
|---|---|---|---|
| DP3 (Ze et al., 2024) | 79.2 | 48.0 | 38.7 |
| **EmbodiedMAE-PC (Ours)** | 79.8 | 76.7 | **68.7** |
| EmbodiedMAE-PC (SparseConv) | **79.9** | **77.6** | 68.2 |

## C SCALING BEHAVIOR WITH LIMITED DATA

To understand the data efficiency and scaling behavior of EmbodiedMAE when trained on reduced dataset sizes, we conduct supplementary experiments utilizing subsets of the DROID-3D dataset. We perform the distillation process for the EmbodiedMAE-Large variant across three dataset scales: 100%, 50%, and 25% of the full DROID-3D dataset. The re-distilled models are evaluated on the LIBERO-Goal benchmark for RGB and RGBD inputs, and on the MetaWorld benchmark for the PC input. The results, summarized in Table 11, demonstrate a minimal and non-significant performance reduction as the training data size is progressively decreased.

Table 11: Ablation study of EmbodiedMAE-Large performance across DROID-3D dataset subsets.

| Task - Configuration | Full Dataset (100%) | Half Dataset (50%) | Quarter Dataset (25%) |
|---|---|---|---|
| LIBERO-Goal-RGB | 91.8 | 91.3 | 90.4 |
| LIBERO-Goal-RGBD | 93.6 | 93.3 | 92.7 |
| MetaWorld-PC | 77.7 | 76.9 | 75.1 |

## D  COMPARISON WITH STATE-OF-THE-ART 3D ESTIMATION MODEL

We evaluate the performance of EmbodiedMAE against models derived from powerful 3D geometric estimation architectures, specifically comparing with VGGT (Wang et al., 2025). VGGT is recognized as a strong baseline, designed to accurately predict geometric maps (such as depth or point clouds) from an RGB image, suggesting strong implicit spatial perception capabilities. While EmbodiedMAE is a multi-modal masked autoencoder focused on representation learning, we assess VGGT's effectiveness as an image representation for policy learning by comparing its performance, as reported by VGGT-DP (Ge et al., 2025). VGGT-DP is a Diffusion Policy that leverages VGGT's intermediate features, demonstrating a best-practice approach for utilizing VGGT in embodied control. The results, presented in Table 12, indicate that despite its strength in geometric estimation, the representation learned by VGGT underperforms EmbodiedMAE significantly across all tested configurations. VGGT-DP achieved an average success rate of 37.4%. EmbodiedMAE consistently demonstrates superior performance in policy learning. The multi-modal EmbodiedMAE-RGBD achieves a high average success rate of 70.8%, showcasing the benefit of pre-training specifically for multi-modal feature fusion and downstream manipulation. This comparison reinforces the finding that representation models explicitly trained on domain-specific embodied data for manipulation, like EmbodiedMAE, provide a superior foundation for control policies compared to features derived from models primarily focused on geometric estimation.

Table 12: Comparison of Policy Performance on a subset of MetaWorld tasks using EmbodiedMAE vs. VGGT representations. Results of VGGT-DP are from its paper, Table 1.

| Representation Model | Sweep-into | Soccer | Shelf-place | Disassemble | Stick-pull | Average |
|---|---|---|---|---|---|---|
| VGGT-DP (Ge et al., 2025) | 44.0 | 30.0 | 10.0 | 55.0 | 48.0 | 37.4 |
| DP + EmbodiedMAE-RGB | 54.7 | 18.7 | 26.0 | 69.3 | 78.0 | 49.3 |
| DP + EmbodiedMAE-RGBD | 81.3 | **88.0** | **86.0** | 32.0 | 66.7 | **70.8** |
| DP + EmbodiedMAE-PC | **80.0** | 32.0 | 32.0 | **88.0** | **86.0** | 63.6 |

## E  INFERENCE LATENCY ANALYSIS

For practical robot deployment, the latency of the representation model is a critical factor. Since the core architecture of EmbodiedMAE is based on Vision Transformers (ViT), the inference speed is primarily determined by the ViT configuration (model size, sequence length, and batch size). We measure the inference latency of all EmbodiedMAE variants on a single NVIDIA GeForce RTX 4090 (24GB) GPU, reporting results for both single-precision floating point (fp32) and BFloat16 (bf16) precision across varying batch sizes and input modality counts. The latency values presented below focus solely on the forward pass time of the EmbodiedMAE encoder.

As shown in Table 13, smaller models like EmbodiedMAE-S and EmbodiedMAE-B maintain low latency (around 16 ms to 17 ms) even when increasing the number of modalities. The larger models, EmbodiedMAE-L and EmbodiedMAE-G, exhibit improved efficiency when switching to bf16 precision, particularly noticeable for the Giant model, where bf16 is essential to avoid Out-of-Memory (OOM) errors at higher batch sizes. The efficient scaling behavior in bf16 makes the EmbodiedMAE-L and EmbodiedMAE-G variants practical for use in high-performance robotics pipelines.

Table 13: Inference Latency (ms) of EmbodiedMAE Variants on a Single RTX 4090 GPU.

| Model (Params) | fp32 Precision | | | fp32 Precision | | |
|---|---|---|---|---|---|---|
| | bs = 1, 1-mod. | bs = 1, 2-mod. | bs = 1, 3-mod. | bs = 8, 1-mod. | bs = 8, 2-mod. | bs = 8, 3-mod. |
| EmbodiedMAE-S (22.1M) | 15.8 | 16.0 | 16.1 | 17.4 | 17.3 | 17.8 |
| EmbodiedMAE-B (87.0M) | 16.7 | 17.1 | 17.3 | 17.7 | 24.5 | 37.8 |
| EmbodiedMAE-L (305.2M) | 29.2 | 31.6 | 33.7 | 74.2 | 76.6 | 122.0 |
| EmbodiedMAE-G (1.1B) | 118.0 | OOM | OOM | OOM | OOM | OOM |
| Model (Params) | bf16 Precision | | | bf16 Precision | | |
| | bs = 1, 1-mod. | bs = 1, 2-mod. | bs = 1, 3-mod. | bs = 8, 1-mod. | bs = 8, 2-mod. | bs = 8, 3-mod. |
| EmbodiedMAE-S (22.1M) | 15.8 | 16.0 | 16.1 | 17.5 | 17.5 | 17.3 |
| EmbodiedMAE-B (87.0M) | 16.7 | 17.0 | 17.2 | 16.5 | 17.4 | 17.5 |
| EmbodiedMAE-L (305.2M) | 29.2 | 31.6 | 31.8 | 31.4 | 36.8 | 46.9 |
| EmbodiedMAE-G (1.1B) | 53.9 | 53.4 | 55.2 | 58.7 | 84.9 | 145.0 |

# F    USE OF LLM

We utilized LLMs as a writing assistance tool during the preparation of this manuscript. The use of LLMs was strictly limited to polishing the text, which included improving grammar, refining sentence structure, and enhancing overall clarity and readability. The core research concepts, methodologies, and conclusions were developed entirely by the authors.

