# OpenReview forum: "EmbodiedMAE: A Unified 3D Multi-Modal Representation for Robot Manipulation"
_ICLR.cc/2026/Conference — Submitted to ICLR 2026_

### Official Review · Reviewer_aDrd · 2025-10-15

**Soundness:** 3
**Presentation:** 3
**Contribution:** 3
**Rating:** 6
**Confidence:** 5

**Summary:**

This paper introduces EmbodiedMAE, a novel framework addressing the challenge of 3D multi-modal representation learning for embodied AI. A key contribution is the introduction of the DROID-3D dataset, which extends the large-scale DROID robot manipulation dataset by incorporating depth maps generated from its stereo camera data via the ZED SDK. The core of the EmbodiedMAE method is a multi-modal Masked Autoencoder (MAE) pre-trained on RGB, depth, and point cloud data to learn a robust representation encoder. To create more efficient models, the authors distill knowledge from a ViT-giant model, producing small, base, and large variants of their encoder. For downstream evaluation, the framework is paired with a compact version of a Robotic Decision Transformer (RDT) as a policy network. Experiments conducted in both simulated and real-world robotic manipulation tasks demonstrate that EmbodiedMAE achieves superior performance, validating its effectiveness for embodied intelligence.

**Strengths:**

- The paper is well-written and clearly structured. The contributions are easy for the reader to understand and appreciate.

- The development of the DROID-3D dataset, which augments an existing large-scale robotics dataset with processed depth information, is a notable contribution. If made publicly available, this dataset will be a valuable asset to the embodied AI and robotics research communities.

- The proposed method is technically sound, addressing the important problem of multi-modal representation learning for embodied AI.

- The potential ability to perform depth2RGB, RGB2Depth, and Re-Color is really interesting.

- This work has a thorough experimental validation, which encompasses both simulation and real-world robotic manipulation tasks. This strengthens the paper’s conclusions, convincingly demonstrating the robustness and practical applicability of the method.

**Weaknesses:**

1. The reliance on the ZED SDK for depth estimation warrants further discussion. Depth data from stereo SDKs can often be noisy, sparse, or contain artifacts, which could significantly impact the quality of the learned representations. The paper would be strengthened by:

    - Clarifying if any specific preprocessing or filtering techniques were applied to the raw depth maps.

    - Discussing the potential impact of depth noise on the model's performance.

    - Exploring or at least acknowledging more robust solutions, such as hybrid methods that combine SDK output with modern AI-based depth completion or denoising networks.

2. The DROID dataset provides rich multi-view information (two third-person, one ego-centric) that appears to be under-leveraged. The manuscript is unclear on which view(s) were used for pre-training and how the point clouds were generated. This raises two key points:

    - It should be explicitly stated whether the point clouds are derived from a single view or fused from multiple views.

    - Leveraging multi-view consistency could be a powerful tool to denoise the depth data or generate more complete 3D point clouds. The current single-view (or ambiguously generated) approach may be a significant limitation.

3. The selection of FPS and a DP3-style encoder for point cloud processing is a potential weakness. These methods, to my knowledge, are outdated and poor in 3D vision. The authors should either justify this architectural choice or, ideally, compare it against more contemporary and powerful alternatives such as those based on sparse convolutions (e.g., Zhu et al., 2024) to demonstrate that the chosen backbone is not a performance bottleneck.

4. The experimental comparisons between EmbodiedMAE and baselines like SPA and DINOV2 are potentially confounded by mismatched model sizes (e.g., comparing a 'Giant' variant to a 'Large' variant). For a fair and convincing evaluation, the authors should:

    - Provide a direct comparison of performance with all methods normalized to the same model size (e.g., all at 'Base' or 'Large' scales).

    - Include a parameter count table for all compared encoder models in different tables.

    - Present a scaling analysis curve (performance vs. model size) for EmbodiedMAE to better characterize its efficiency and scalability.

5. The core motivation for simultaneously pre-training on three modalities (RGB, depth, and point cloud) is not fully substantiated. It is unclear if this tri-modal approach offers a true synergistic benefit. The paper would be much more compelling with rigorous ablation studies that isolate the contribution of each modality. For instance:

    - How does the full model compare to variants pre-trained on simpler combinations, such as RGB-only, Depth-only, or RGB+Depth?

    - If the downstream policy primarily consumes point clouds, is the point cloud branch of EmbodiedMAE superior to state-of-the-art encoders pre-trained exclusively on point cloud data? Without these ablations, the necessity of the proposed multi-modal complexity remains an open question.

[1] Haoyi Zhu, Yating Wang, Di Huang, Weicai Ye, Wanli Ouyang, and Tong He. Point cloud matters: Rethinking the impact of different observation spaces on robot learning. In The Thirty-eight Conference on Neural Information Processing Systems Datasets and Benchmarks Track, NIPS, 2024.

**Questions:**

See the weakness section

---

> ### Author Response · Authors · 2025-11-21
>
> ### 1. Depth Pre-processing
>
> **Depth:** Although the ZED camera is a stereo vision camera which often does not provide high-quality depth data, the ZED SDK itself encapsulates AI-enhanced models and temporal data fusion. Thus, using the ZED SDK directly yields processed high-quality depth maps. We attribute the contribution of enhancing depth map quality to the **ZED SDK** (https://www.stereolabs.com/docs/depth-sensing/using-depth), while our contribution is setting the **ULTRA quality depth enhancement** and expending compute to acquire the dataset, thereby avoiding other researchers from having to repeatedly expend significant compute to obtain the 3D DROID dataset.
>
> **Point Cloud:** After obtaining the high-quality depth map, we use the camera intrinsics to derive the **Point Map**, then use **voxel downsampling** to acquire an initial point cloud with uniform spatial distribution, followed by **bounding clip** to remove extremely distant points, **Radius Outlier Removal** to eliminate scattered outliers, and finally **FPS** to downsample to 8192 points to balance quality and storage space.
>
> ### 2. About Multi-view Information in DROID Dataset
>
> We **actually did not utilize multi-view** information but instead treated every single image as an **independent data point**. This is because we did not want the representation model to rely on multi-view relationships, as that would make the model difficult to use in a single-view setting. Correspondingly, our point cloud data is also derived from **single-view depth**. Although the single-view point cloud is incomplete, for many robotics scenarios, the robot only sees a partial, single-sided point cloud. Nevertheless, the data organization in the DROID-3D dataset is still based on the multi-view structure, allowing users to easily leverage multi-view information or merge multi-view point clouds into a complete one using camera extrinsics, if needed.
>
> ### 3. Design Choice of Point Cloud Encoder
>
> - **Justification:** When designing the network architecture, we aimed to concentrate all feature extraction within the Transformer backbone. Consequently, we adopted super light-weighted structures for the pre-encoding of all three modalities. For example, the patchifier for RGB and Depth was implemented as a non-overlapping Conv2D (a common choice). Similarly, we desired a simple "patchifying" for the Point Cloud modality, which led to the combination of **FPS + KNN + DP3 Encoder**. This structure primarily serves a **tokenization** role, while feature extraction and cross-modal fusion are expected to be performed in the subsequent Transformer encoder.
> - **Sparse Convolution Alternative:** Inspired by your suggestion, we are also curious whether using a more powerful PC encoder would enhance model performance. We replace the PC encoder of EmbodiedMAE-Large with **SparseConv3d** (from https://github.com/traveller59/spconv/blob/master/spconv/pytorch/conv.py) and further distill the model from EmbodiedMAE-Giant (we omitted this part of the modality encoder distillation loss due to the change in the modality encoder). We conduct supplementary experiments on the ****MetaWorld benchmark and the xArm platform (Please see EmbodiedMAE-PC-SparseConv below ) and observe **no significant performance change** on both. We believe this demonstrates that feature extraction indeed occurs in the Transformer backbone.
> - **Pre-processing Techniques to Improve Performance:** Inspired by Reviewer **vzfu**, we enhance the L515 camera data used on the xArm platform. This includes applying **radius outlier removal** to the depth map, using **CrocoV2-Stereo generated metric depth as guidance for a guided filter**, converting the result into a point cloud, applying **voxel downsampling**, **bounding clip**, and finally **FPS**. After implementing this on the xArm platform (Please see EmbodiedMAE-PC-enhancedPCquality and DP3-enhancedPCquality), we find that **EmbodiedMAE-PC** shows a significant **performance** **improvement**, whereas DP3 did not. This suggests that improved PC quality effectively enhances the modality representation for **EmbodiedMAE-PC**, while DP3, which lacks a pre-training process, achieves its standard performance as long as the training and inference data distributions remain consistent.
>
> |  | MetaWorld Easy (18) | MetaWorld Medium (9) | MetaWorld Very Hard (3) | xArm Pick&Place | xArm Pot | xArm Pour | xArm Moka | xArm average |
> | --- | --- | --- | --- | --- | --- | --- | --- | --- |
> | EmbodiedMAE-PC | 79.8 | 76.7 | 68.7 | 73.3 | 75.0 | 80.0 | 80.0 | 77.1 |
> | EmbodiedMAE-PC (SparseConv) | 79.9 | 77.6 | 68.2 | 71.6 | 85.0 | 80.0 | 80.0 | 79.2 |
> | EmbodiedMAE-PC (enhanced PC quality) | - | - | - | 93.3 | 85.0 | 80.0 | 70.0 | 82.1 |
> | DP3 | 79.2 | 48.0 | 38.7 | 71.7 | 55.0 | 90.0 | 70.0 | 71.7 |
> | DP3 (enhanced PC quality) | - | - | - | 76.7 | 55.0 | 90.0 | 70.0 | 72.9 |

---

> ### Author Response · Authors · 2025-11-21
>
> ### 4. Comparison with PonderV2
>
> Thank you for your suggestion. In our initial PC experiments, we only compared with DP3 because common point cloud VFM training data is mainly at the outdoor/indoor scene or object level, leading to a significant gap with tabletop manipulation. To better showcase our model's advantages, we add a comparison with **PonderV2** (https://github.com/OpenGVLab/PonderV2) on the xArm platform. We observe a **performance drop** even compared to DP3. This is likely due to the large data distribution gap, preventing the pre-trained representation from functioning effectively.
>
> ### 5. Comparison Under the Same Model Size & Parameter Count
>
> **Comparison under the same model size:** In fact, all current comparisons are conducted under the **same model size**. In the paper, we used *italics* to emphasize that **EmbodiedMAE without a suffix refers to the Large-size model**, and other baselines are also Large-size models. We ensure a **fair comparison** regarding model size. To prevent oversight, we will subsequently replace all instances of EmbodiedMAE with **EmbodiedMAE-L**.
>
> **Parameter Count Table:** Yes, we will add a table in the paper showing the parameter counts.
>
> | EmbodiedMAE-S | EmbodiedMAE-B | EmbodiedMAE-L | EmbodiedMAE-G |
> | --- | --- | --- | --- |
> | 22.1M | 87.0M | 305.2M | 1.1B |
>
> **Scaling Analysis Curve:** We have a scaling analysis curve. **Figure 6, second row** compares the learning curves of EmbodiedMAE across **four scales**, comprehensively contrasting (performance vs. model size).
>
> ### 6. About Tri-modal Motivation
>
> We want to reiterate that our motivation for choosing the **RGB-Depth-PC tri-modal** approach is not because we believe this combination necessarily *maximizes* the representation model's performance, but because these three modalities are the **most commonly used** in real-world robotics scenarios (RGB camera, stereo vision camera, LiDAR camera, and other sensors). For engineering application considerations, we want one representation model to be able to **handle all these working conditions**. Furthermore, due to the design of the mask strategy, the model is trained to process **RGB-only, Depth-Only, and RGB+Depth** situations, which can be considered sub-tasks of our pre-training.

---

> > ### Author Response · Authors · 2025-11-25
> > **Table of comparison with PonderV2 PointCloud VLM**
> >
> > ### 4. Comparison with PonderV2
> >
> > |  | xArm Pick&Place | xArm Pot | xArm Pour | xArm Moka | xArm average |
> > | --- | --- | --- | --- | --- | --- |
> > | EmbodiedMAE-PC | 73.3 | 75.0 | 80.0 | 80.0 | 77.1 |
> > | DP3 | 71.7 | 55.0 | 90.0 | 70.0 | 71.7 |
> > | PonderV2 | 66.7 | 40.0 | 70.0 | 40.0 | 54.1 |

---

> > > ### Comment · Reviewer_aDrd · 2025-11-25
> > >
> > > I thank the authors for their detailed response and additional experiments. Given that the rebuttal has effectively addressed my primary reservations, I am raising my score to 8 "accept, good paper (poster)".

---

### Official Review · Reviewer_vzfu · 2025-10-31

**Soundness:** 3
**Presentation:** 3
**Contribution:** 3
**Rating:** 6
**Confidence:** 4

**Summary:**

This paper presents EmbodiedMAE, a novel framework for learning unified 3D multi-modal representations tailored for robot manipulation. The authors address two critical issues in current embodied AI research: the domain gap between standard Vision Foundation Model (VFM) training data and robotic tasks, and the lack of effective architectures for incorporating 3D perception. Their solution involves creating DROID-3D, an enhanced version of the DROID dataset with high-quality depth maps and point clouds, and developing a multi-modal masked autoencoder that learns joint representations across RGB, depth, and point cloud modalities through stochastic masking and cross-modal fusion. Extensive evaluation across 70 simulation tasks and 20 real-world tasks demonstrates that EmbodiedMAE outperforms state-of-the-art VFMs in both training efficiency and final performance, while showing favorable scaling properties and enabling effective policy learning from 3D inputs.

**Strengths:**

1. This work provides an end-to-end solution addressing both data scarcity (via DROID-3D) and architectural limitations (via EmbodiedMAE), offering substantial value to the research community.

2. The use of ZED SDK for generating metric depth maps and point clouds represents a significant improvement over commonly used estimated depth methods, providing temporally consistent 3D data that is crucial for robotic manipulation.

3. The combination of modality-agnostic stochastic masking (using Dirichlet distribution) with explicit cross-modal fusion in the decoder is elegant and effectively encourages learning aligned representations across different modalities.

4. The experimental design is particularly strong, covering diverse simulation environments and real-world platforms, with comprehensive comparisons against relevant VFM baselines.

5. The distillation approach from Giant to smaller variants makes the technology accessible for practical applications while maintaining strong performance.

**Weaknesses:**

1. While the paper demonstrates strong performance metrics, it lacks discussion of inference latency for the different EmbodiedMAE variants. For real-world robotic deployment where real-time performance is critical, understanding the latency characteristics on target hardware would help assess practical applicability.

2. The observation that point-cloud-only policies underperform RGB-only inputs is noted but not thoroughly analyzed. The paper would benefit from exploring whether alternative point cloud processing techniques could improve PC-only performance.

3. The ablation study focuses mainly on masking ratios but provides limited analysis of the Dirichlet distribution parameter α. Understanding how different α values affect learning across various task types would offer deeper insights into optimal masking strategies for multi-modal representation learning.

**Questions:**

1. What is the inference latency of different EmbodiedMAE variants (particularly Base and Large) on common robotic hardware platforms? How does this compare to baseline models in terms of the latency-performance trade-off?

2. Given the underperformance of PC-only policies, did the authors experiment with alternative point cloud encoders or pre-processing techniques to improve robustness to sensor noise? What modifications might help bridge this performance gap?

3. How sensitive are the pre-training results to the Dirichlet parameter α? Have the authors experimented with different α values to understand its impact on cross-modal learning efficiency?

---

> ### Author Response · Authors · 2025-11-21
>
> ### 1. Inference Latency
>
> Policy latency in robotics deployment involves multiple factors. As we focus on the representation model, here we provide the **inference latency** for all **EmbodiedMAE variants**, tested on a single 24G RTX 4090 GPU. Since our main baselines are ViT structures, the inference latency is almost entirely dependent on the **ViT configuration**. (OOM here means out-of-memory case)
>
> | **Latency (ms)** | fp32,bs=1,1-modality | fp32,bs=1,2-modality | fp32,bs=1,3-modality | fp32,bs=8,1-modality | fp32,bs=8,2-modality | fp32,bs=8,3-modality |
> | --- | --- | --- | --- | --- | --- | --- |
> | EmbodiedMAE-S (22.1M) | 15.8 | 16 | 16.1 | 17.4 | 17.3 | 17.8 |
> | EmbodiedMAE-B (87.0M) | 16.7 | 17.1 | 17.3 | 17.7 | 24.5 | 37.8 |
> | EmbodiedMAE-L (305.2M) | 29.2 | 31.6 | 33.7 | 74.2 | 76.6 | 122 |
> | EmbodiedMAE-G (1.1B) | 118 | OOM | OOM | OOM | OOM | OOM |
> ||  bf16,bs=1,1-modality | bf16,bs=1,2-modality | bf16,bs=1,3-modality | bf16,bs=8,1-modality | bf16,bs=8,2-modality | bf16,bs=8,3-modality |
> | EmbodiedMAE-S (22.1M) | 15.8 | 16 | 16.1 | 17.5 | 17.5 | 17.3 |
> | EmbodiedMAE-B (87.0M) | 16.7 | 17 | 17.2 | 16.5 | 17.4 | 17.5 |
> | EmbodiedMAE-L (305.2M) | 29.2 | 31.6 | 31.8 | 31.4 | 36.8 | 46.9 |
> | EmbodiedMAE-G (1.1B) | 53.9 | 53.4 | 55.2 | 58.7 | 84.9 | 145 |
>
> ### 2. Design Choice of Point Cloud Encoder
>
> - **Justification:** When designing the network architecture, we aimed to concentrate all feature extraction within the Transformer backbone. Consequently, we adopted super light-weighted structures for the pre-encoding of all three modalities. For example, the patchifier for RGB and Depth was implemented as a non-overlapping Conv2D (a common choice). Similarly, we desired a simple "patchifying" for the Point Cloud modality, which led to the combination of **FPS + KNN + DP3 Encoder**. This structure primarily serves a **tokenization** role, while feature extraction and cross-modal fusion are expected to be performed in the subsequent Transformer encoder.
> - **Sparse Convolution Alternative:** Inspired by Reviewer **aDrd**, we are also curious whether using a more powerful PC encoder would enhance model performance. We replace the PC encoder of EmbodiedMAE-Large with **SparseConv3d** (from https://github.com/traveller59/spconv/blob/master/spconv/pytorch/conv.py) and further distill the model from EmbodiedMAE-Giant (we omitted this part of the modality encoder distillation loss due to the change in the modality encoder). We conduct supplementary experiments on the ****MetaWorld benchmark and the xArm platform (Please see EmbodiedMAE-PC-SparseConv below ) and observe **no significant performance change** on both. We believe this demonstrates that feature extraction indeed occurs in the Transformer backbone.
> - **Pre-processing Techniques to Improve Performance:** Inspired by your suggestion, we enhance the L515 camera data used on the xArm platform. This includes applying **radius outlier removal** to the depth map, using **CrocoV2-Stereo generated metric depth as guidance for a guided filter**, converting the result into a point cloud, applying **voxel downsampling**, **bounding clip**, and finally **FPS**. After implementing this on the xArm platform (Please see EmbodiedMAE-PC-enhancedPCquality and DP3-enhancedPCquality), we find that **EmbodiedMAE-PC** shows a significant **performance** **improvement**, whereas DP3 did not. This suggests that improved PC quality effectively enhances the modality representation for **EmbodiedMAE-PC**, while DP3, which lacks a pre-training process, achieves its standard performance as long as the training and inference data distributions remain consistent.
>
> |  | MetaWorld Easy (18) | MetaWorld Medium (9) | MetaWorld Very Hard (3) | xArm Pick&Place | xArm Pot | xArm Pour | xArm Moka | xArm average |
> | --- | --- | --- | --- | --- | --- | --- | --- | --- |
> | EmbodiedMAE-PC | 79.8 | 76.7 | 68.7 | 73.3 | 75.0 | 80.0 | 80.0 | 77.1 |
> | EmbodiedMAE-PC (SparseConv) | 79.9 | 77.6 | 68.2 | 71.6 | 85.0 | 80.0 | 80.0 | 79.2 |
> | EmbodiedMAE-PC (enhanced PC quality) | - | - | - | 93.3 | 85.0 | 80.0 | 70.0 | 82.1 |
> | DP3 | 79.2 | 48.0 | 38.7 | 71.7 | 55.0 | 90.0 | 70.0 | 71.7 |
> | DP3 (enhanced PC quality) | - | - | - | 76.7 | 55.0 | 90.0 | 70.0 | 72.9 |
>
> ### 3. Dirichlet Distribution Parameter
>
> We must regretfully state that, due to the **limited time** of the rebuttal period, it is **truly impossible** for us to perform an ablation study on the Dirichlet distribution parameter, as this involves a significant number of pre-training experiments. However, we can justify the current choice of **$\alpha=1.0$** as a **balanced scheme for all modalities** that has been validated by previous works. Logically, we believe that adjusting the masking ratio proportions of different modalities will **not have a significant performance impact** in non-extreme settings.

---

### Official Review · Reviewer_RVAu · 2025-10-31

**Soundness:** 3
**Presentation:** 3
**Contribution:** 3
**Rating:** 4
**Confidence:** 4

**Summary:**

The paper introduces EmbodiedMAE, a unified 3D multi-modal masked autoencoder designed to improve sensory representation for robot manipulation by jointly learning from RGB, depth, and point cloud inputs. To overcome domain gaps and low-quality 3D data in existing embodied datasets, the authors build DROID-3D, augmenting the DROID dataset with high-fidelity metric depth maps and point clouds via ZED SDK processing. EmbodiedMAE uses stochastic modality masking and cross-modal fusion to learn spatially grounded representations, and a ViT-Giant model is pre-trained then distilled into smaller variants. Evaluated across 70 simulation tasks and 20 real-world robot manipulation tasks on two platforms, the method consistently outperforms state-of-the-art vision foundation models, scales effectively with model size, and proves especially strong when leveraging 3D sensing, demonstrating improved spatial perception and manipulation precision in both simulation and real-world settings.

**Strengths:**

- The paper is clearly written and easy to follow, with technical concepts presented in a coherent and accessible manner.
- The experimental setup is well-structured and comprehensive, demonstrating careful design and thorough evaluation across diverse benchmarks.
- The analysis in Section 3.2 presents intriguing and insightful experimental designs that effectively validate the model’s cross-modal learning capabilities.
- The work makes a meaningful contribution to the open-source community by releasing both the DROID-3D dataset and the EmbodiedMAE model implementation.
- The results compellingly demonstrate the benefits of developing a dedicated vision foundation model for embodied AI.

**Weaknesses:**

- The paper does not clearly justify the choice of MAE-based pre-training over alternative paradigms such as CLIP-style contrastive learning or DINO-style self-distillation. This decision is central to the method’s novelty, yet MAE is introduced abruptly (e.g., L48) without sufficient motivation or discussion of trade-offs. A deeper explanation of why MAE is particularly suitable for embodied 3D perception — and why contrastive or language-conditioned methods may be less effective — would significantly strengthen the narrative. In addition, ablations comparing MAE to CLIP-like or DINO-like pre-training objectives would provide stronger empirical evidence for this design choice.

- Considering the submission timeline, VGGT (CVPR 2025) and other emerging 3D transformer backbones are not included as baselines. While understandable, this omission makes it difficult to assess the competitiveness of EmbodiedMAE against the latest 3D perception architectures. A discussion or retrospective comparison to VGGT-style models would help contextualize performance relative to the rapidly evolving 3D VFM landscape.

**Questions:**

- L209, there appears to be a formatting issue with the punctuation — specifically, a period placed before the citation: ... . (He et al., 2022). Please revise to maintain standard citation formatting.

- The paper notes that the predicted RGB images appear smoothed due to the use of L2 reconstruction loss, yet the predicted depth images do not exhibit similar smoothing artifacts. Could the authors provide further clarification on this discrepancy? In particular, why does L2-loss cause noticeable smoothing for RGB reconstruction but not for depth prediction?

---

> ### Author Response · Authors · 2025-11-21
>
> ### 1. Justification of MAE-based Pre-training
>
> **Why not CLIP-style pre-training?** We believe that the CLIP training paradigm leans toward a broad understanding of modalities. The contrastive learning loss does not explicitly force the model to understand patch-level detailed features but rather aims to connect semantic relations between multiple modalities, making it more suitable for image-language representation learning. If we were to apply CLIP contrastive learning on RGB-Depth-PC triplets, since each triplet visually represents the same scene, the model could easily learn a shortcut by maximizing the CLIP score for triplets with identical scenes. This would not be conducive to effective modality fusion.
>
> **Why not DINO-style pre-training?** Although the DINO series represents state-of-the-art self-supervised representation learning, their training loss is composed of a combination of **multiple, carefully designed losses**, naturally suitable for **single-image** tasks. Many of these losses do not support, or cannot be simply adapted for, learning across three modalities.
>
> **Why MAE-style pre-training?** MAE is technically mature and offers stable training. The idea of using partial modalities to **reconstruct and complete** the masked ones is highly appealing, as it **explicitly forces feature fusion** between modalities. As we show in Figure 3, the model can achieve **modality conversion and Re-coloring**, which are visualizations that demonstrate the superiority of MAE in modality understanding.
>
> ### 2. Comparison with VGGT
>
> From a design philosophy perspective, **VGGT** is designed to estimate camera/depth/point maps from an RGB image. It is **neither a representation model** nor can it accept the **RGB-Depth-PC tri-modal input** common in robotics applications, unlike EmbodiedMAE. Furthermore, EmbodiedMAE was **not designed as a modality estimation model**, just like MAEs are not intended for image generation.
>
> Nevertheless, we can still use the intermediate features of VGGT as a policy representation. **VGGT-DP (**https://arxiv.org/abs/2509.18778**)** is a Diffusion Policy that uses VGGT as an image representation. It explores the best practices for combining VGGT and Diffusion Policy and, like EmbodiedMAE, uses the MetaWorld benchmark. We compare the results of VGGT-DP (from Table 1 in the paper) and EmbodiedMAE and find that **VGGT does not perform outstandingly as a representation model.**
>
> | Task Name | Sweep-into | Soccer | Shelf-place | Disassemble | Stick-pull | Average |
> | --- | --- | --- | --- | --- | --- | --- |
> | Diffusion Policy + EmbodiedMAE-RGB | 54.7 | 18.7 | 26.0 | 69.3 | 78.0 | 49.3 |
> | Diffusion Policy + EmbodiedMAE-RGBD | 81.3 | 88.0 | 86.0 | 32.0 | 66.7 | 70.8 |
> | Diffusion Policy + EmbodiedMAE-PC | 80.0 | 32.0 | 32.0 | 88.0 | 86.0 | 63.6 |
> | VGGT-DP | 44.0 | 30.0 | 10.0 | 55.0 | 48.0 | 37.4 |
>
> ### 3. L209 Citation Formatting
>
> Sorry for the formatting issue. We will correct it immediately.
>
> ### 4. About Smoothing Artifacts
>
> We actually observed smoothing artifacts in **all modalities** (RGB-Depth-PC). As you can find in Figure 3, all predicted Depth maps exhibit **smoother boundaries and fewer sharp discontinuities**. This situation is not exclusive to RGB. Such smoothing artifacts are common in all MAE models and even existed in many early generative models, often requiring adversarial training losses for resolution. However, given that **EmbodiedMAE's goal is to be a representation model**, we do not consider this a critical issue.

---

> > ### Comment · Reviewer_RVAu · 2025-11-22
> > **Thank you for the detailed rebuttal and the additional experiments**
> >
> > Thank you for the detailed rebuttal and the additional experiments. Several concerns are clarified, and I appreciate the effort. I would like to follow up on Weakness 2 regarding VGGT.
> >
> > In your response, you argue that VGGT is “not a representation model” because it is designed for 3D prediction rather than multimodal representation learning. However, just curious, this raises an important conceptual question:
> >
> > Follow-up Question
> >
> > How do you define what counts as a “representation model”?
> > Specifically:
> >
> > - Is a model not a representation model simply because it was not trained with representation learning methods: e.g., MAE/contrastive/self-distillation objectives?
> >
> > - If a model is trained for a geometric task but produces transferable intermediate features, does it still not qualify as a representation backbone?
> >
> > More broadly, what criteria do you use to distinguish a “representation model” from a “3D vision model”?
> >
> > This clarification matters because many widely used backbones (e.g.,  depth-prediction encoders, sam) were not originally trained as “representation models,” yet their features transfer effectively. Thus, I encourage the authors to more clearly articulate their definition and rationale, rather than relying on the intended purpose of the original training task.

---

> > > ### Author Response · Authors · 2025-11-25
> > >
> > > Thanks for your reply. We are very pleased to follow up on this conceptual question with you.
> > >
> > > From my perspective—including my understanding gleaned from tutorials, surveys, and papers—a clear, unified definition for "representation models" is indeed lacking. There is only a vague, shared expectation: that the learned representations should help improve the training and performance of a majority of downstream tasks (https://arxiv.org/pdf/2307.13721, https://arxiv.org/pdf/1206.5538). Based on this view, here is my personal perspective on your questions:
> > >
> > > ### 1. "Is a model not a representation model simply because it was not trained with representation learning methods?"
> > >
> > > Basically, **yes**, but the reasoning extends beyond a simple classification of the loss function. I believe the representations we use are the result of clustering during training.
> > >
> > > * When we use a prediction loss, it provides a more explicit directive to the clustering process: the model can discard any unnecessary information required only to achieve a highly accurate prediction.
> > > * In contrast, when using representation learning methods, the clustering directive is milder, essentially requiring the model to retain more information from the original data, resulting in less aggressive compression.
> > >
> > > Therefore, I prefer to refer to models trained with non-predictive losses as "representation models" because their features are logically applicable to a wider range of downstream tasks.
> > >
> > > ### 2. "If a model is trained for a geometric task but produces transferable intermediate features, does it still not qualify as a representation backbone?"
> > >
> > > I believe it **is qualified**. Although I would not categorize it as a "representation model," it can certainly be used as a representation backbone, and it is highly likely to perform better on specific downstream tasks. For instance, the intermediate features of VGGT should logically exhibit superior performance on downstream tasks related to monocular spatial awareness.
> > >
> > > ### Returning to Our Work
> > >
> > > To bring this back to our work: I still feel that using the intermediate features of predictive models like VGGT was not an immediate choice for a baseline (though of course, it *can* and *should* serve as one), but its excellent monocular depth estimation capabilities certainly made me highly anticipate its performance on robot manipulation tasks. Sadly, our current experimental results show it does not possess an advantage in this domain. I would suggest that EmbodiedMAE's key advantage (compared to VGGT) lies in its ability to accept any combination of RGB, Depth, and PC inputs, which allows it to adapt to a greater variety of robot configurations.
> > >
> > > Thank you again for your insightful reply. I look forward to hearing your perspective on this question and further discussion.

---

> > > > ### Comment · Reviewer_RVAu · 2025-11-25
> > > > **Thank you for your on time reply**
> > > >
> > > > Thank you for discussing this question and for the detailed rebuttal. Your clarifications on the definition of representation models, the role of non-predictive losses, and the comparison with VGGT address most of my concerns. Based on this, I am considering raising my score. Please continue improving the clarity and flow of the writing.

---

> > > > > ### Author Response · Authors · 2025-11-25
> > > > >
> > > > > Thank you for your consideration! The paper is continuously being optimized, and all explanations and experiments supplemented during this rebuttal process will be formatted and integrated into the paper, regardless of the final acceptance decision.

---

### Official Review · Reviewer_Z4rL · 2025-11-01

**Soundness:** 3
**Presentation:** 4
**Contribution:** 3
**Rating:** 4
**Confidence:** 4

**Summary:**

This paper introduces EmbodiedMAE, a unified 3D multi-modal representation learning framework for robot manipulation. The authors first create DROID-3D, a large-scale, high-quality 3D dataset by enhancing the DROID dataset with temporally consistent RGB, depth, and point cloud data. They then propose a multi-modal Masked Autoencoder (MAE) architecture that uses stochastic masking and cross-modal fusion for pre-training. The model, which is initialized with DINOv2 weights and uses a knowledge distillation strategy, consistently outperforms state-of-the-art vision foundation models (VFMs) across 70 simulation tasks (LIBERO, MetaWorld) and 20 real-world tasks on two different robot platforms (SO100 and xArm). The core contribution is a robust VFM that effectively leverages 3D spatial information for precise embodied AI tasks.

**Strengths:**

- The paper is exceptionally well-written and clear. The methodology is easy to follow, from the data preparation for DROID-3D to the detailed explanation of the encoder, multi-modal decoder, and distillation process.
- It provides a robust and scalable method for effectively utilizing 3D inputs, which are crucial for precise manipulation but often degrade performance in prior VFM adaptation attempts.
- The DROID-3D dataset is a valuable public resource, and the demonstration of superior performance on low-cost (SO100) and high-performance (xArm) robots proves the model's practical utility and generalization power.

**Weaknesses:**

- The core contribution is a unified 3D model, yet the real-world results show the Point Cloud (PC) policies (EmbodiedMAE-PC) significantly underperform the RGB-only and RGBD variants on the xArm platform. This contradicts the paper's goal of effective 3D fusion. The paper attributes this to sensor noise, but this suggests the DP3 encoder or the PC representation itself is not robust enough. A more thorough analysis or ablation comparing different PC encoders is necessary to validate the PC pipeline choice.

**Questions:**

- The core results rely on the RDT diffusion policy. The ACT ablation (Table 2) is currently limited to the RGB-only setting. Please extend the ACT policy ablation to include the EmbodiedMAE-RGBD and EmbodiedMAE-PC variants on the LIBERO-Goal benchmark. This is crucial to demonstrate that the superior performance of the 3D features is independent of the RDT policy type.
- The paper notes processing the complete 76K trajectories of DROID (L. 138). Did the authors test the impact of using only a subset of DROID-3D (e.g., 1/2 or 1/4 size) on the final policy learning curve? This would quantify the direct value of the sheer scale of the DROID-3D dataset, which is currently only implied.

---

> ### Author Response · Authors · 2025-11-21
>
> ### 1. Design Choice of Point Cloud Encoder
>
> - **Justification:** When designing the network architecture, we aimed to concentrate all feature extraction within the Transformer backbone. Consequently, we adopted super light-weighted structures for the pre-encoding of all three modalities. For example, the patchifier for RGB and Depth was implemented as a non-overlapping Conv2D (a common choice). Similarly, we desired a simple "patchifying" for the Point Cloud modality, which led to the combination of **FPS + KNN + DP3 Encoder**. This structure primarily serves a **tokenization** role, while feature extraction and cross-modal fusion are expected to be performed in the subsequent Transformer encoder.
> - **Sparse Convolution Alternative:** Inspired by Reviewer **aDrd**, we are also curious whether using a more powerful PC encoder would enhance model performance. We replace the PC encoder of EmbodiedMAE-Large with **SparseConv3d** (from https://github.com/traveller59/spconv/blob/master/spconv/pytorch/conv.py) and further distill the model from EmbodiedMAE-Giant (we omitted this part of the modality encoder distillation loss due to the change in the modality encoder). We conduct supplementary experiments on the ****MetaWorld benchmark and the xArm platform (Please see EmbodiedMAE-PC-SparseConv below ) and observe **no significant performance change** on both. We believe this demonstrates that feature extraction indeed occurs in the Transformer backbone.
> - **Pre-processing Techniques to Improve Performance:** Inspired by Reviewer **vzfu**, we enhance the L515 camera data used on the xArm platform. This includes applying **radius outlier removal** to the depth map, using **CrocoV2-Stereo generated metric depth as guidance for a guided filter**, converting the result into a point cloud, applying **voxel downsampling**, **bounding clip**, and finally **FPS**. After implementing this on the xArm platform (Please see EmbodiedMAE-PC-enhancedPCquality and DP3-enhancedPCquality), we find that **EmbodiedMAE-PC** shows a significant **performance** **improvement**, whereas DP3 did not. This suggests that improved PC quality effectively enhances the modality representation for **EmbodiedMAE-PC**, while DP3, which lacks a pre-training process, achieves its standard performance as long as the training and inference data distributions remain consistent.
>
> |  | MetaWorld Easy (18) | MetaWorld Medium (9) | MetaWorld Very Hard (3) | xArm Pick&Place | xArm Pot | xArm Pour | xArm Moka | xArm average |
> | --- | --- | --- | --- | --- | --- | --- | --- | --- |
> | EmbodiedMAE-PC | 79.8 | 76.7 | 68.7 | 73.3 | 75.0 | 80.0 | 80.0 | 77.1 |
> | EmbodiedMAE-PC (SparseConv) | 79.9 | 77.6 | 68.2 | 71.6 | 85.0 | 80.0 | 80.0 | 79.2 |
> | EmbodiedMAE-PC (enhanced PC quality) | - | - | - | 93.3 | 85.0 | 80.0 | 70.0 | 82.1 |
> | DP3 | 79.2 | 48.0 | 38.7 | 71.7 | 55.0 | 90.0 | 70.0 | 71.7 |
> | DP3 (enhanced PC quality) | - | - | - | 76.7 | 55.0 | 90.0 | 70.0 | 72.9 |
>
> ### 2. Expanded ACT Ablation
>
> To align with the experimental setting of the RDT diffusion policy, we supplement experiments for **EmbodiedMAE-RGBD** and **DINOv2-RGBD** on **LIBERO-Goal**, and **EmbodiedMAE-PC** on **MetaWorld** (we found that using the point cloud modality on LIBERO leads to extremely slow inference speed, making it impossible to complete within the limited rebuttal timeframe). The experimental results indicate that **EmbodiedMAE** using 3D modalities can still effectively **improve the performance** of transformer-based policies.
>
> | LIBERO-Goal | EmbodiedMAE-L-RGB | DINOv2-L-RGB | SPA | EmbodiedMAE-L-RGBD | DINOv2-L-RGBD |
> | --- | --- | --- | --- | --- | --- |
> | ACT Policy | 83.7 | 76.3 | 82.5 | 90.8 | 82.2 |
>
> |  | MetaWorld Easy (18) | MetaWorld Medium (9) | MetaWorld Very Hard (3) |
> | --- | --- | --- | --- |
> | Diffusion Policy + EmbodiedMAE-PC | 79.8 | 76.7 | 68.7 |
> | DP3 | 79.2 | 48.0 | 38.7 |
> | ACT Policy + EmbodiedMAE-PC | 80.0 | 64.4 | 56.2 |
> | ACT Policy + DP3 Encoder | 78.8 | 42.7 | 33.1 |
>
> ### 3. Training on Subset of DROID-3D
>
> This experiment requires a long training time and is currently not yet complete. We will supplement the results and provide feedback to you as soon as they are finalized.

---

> ### Author Response · Authors · 2025-11-25
>
> Thank you for your patience. Based on your suggestion, we have supplemented experiments on EmbodiedMAE trained on a subset of the DROID dataset (full, $1/2$, and $1/4$ scale).
>
> Due to the limited rebuttal time, we were unable to complete the re-pre-training of the Giant model. As an alternative, we re-distilled the EmbodiedMAE-Large model across the three dataset scales. After completing the training, we re-evaluated the RGB and RGBD settings on LIBERO-Goal and the PC setting on MetaWorld (as previously noted, the PC setting on LIBERO runs too slowly). The results are shown in the table below:
>
> | Task - Config / Dataset | 100% | 50% | 25% |
> | :--- | :---: | :---: | :---: |
> | LIBERO-Goal-RGB | 91.8 | 91.3 | 90.4 |
> | LIBERO-Goal-RGBD | 93.6 | 93.3 | 92.7 |
> | MetaWorld-PC | 77.7 | 76.9 | 75.1 |
>
> We observe a minimal, non-significant performance drop as the dataset size is reduced. This robustness might be attributed to the following reasons:
>
> * (a) The presence of the teacher distillation loss during training helps stabilize learning.
> * (b) The ViT backbone parameters were initialized from DINOv2, inherently providing a certain level of visual understanding capability.
> * (c) The DROID dataset is already sufficiently large to support the training. With approximately 27.6M data points per modality, the three modalities (+DROID-3D) combined constitute around 80M data points, which is substantial enough to convey universal robot manipulation information.

---

### Author Response · Authors · 2025-11-27
**Revision and Global Response**

Dear Reviewers,

Thank you for your valuable participation in the revision of our work. We have compiled all the supplementary experiments and explanations from the rebuttal phase and incorporated them into the revised manuscript.

The revision includes the following additions:

1.  **Fixed Citation Formatting Issue (Page 4, L209).**
2.  **Detailed Analysis of Point Cloud Modality (Page 19, Appendix B):** This section includes a comparison with another PC visual foundation model (PonderV2), a comparison using a sparse convolutional network as the PC patchifier, and an analysis of constructing a point cloud data pre-processing pipeline to enhance sensor data quality. Generally speaking, we found that PC VFMs trained on scene-level data do not perform well on robot manipulation tasks, that using a more complex network to replace the current DP3 Encoder as the PC patchifier offered no significant gain, and that enhancing the quality of sensor point cloud data significantly improves real-robot performance.
3.  **Expanded ACT Ablation (Page 9, Section 3.5):** We have expanded the performance evaluation of ACT using EmbodiedMAE with various multimodal inputs, where EmbodiedMAE consistently demonstrates a significant advantage.
4.  **Training Results on Different Dataset Subsets (100%, 50%, 25%) (Page 20, Appendix C).**
5.  **Comparison with VGGT (Page 21, Appendix D):** EmbodiedMAE shows a significant advantage over VGGT in robot manipulation tasks.
6.  **Inference Latency Analysis (Page 21, Appendix E).**

We sincerely hope that this revised version and our comprehensive responses to your comments effectively address your concerns. Should you have any further questions or wish to discuss anything additional, please feel free to add a new comment. Thank you once again for your effort and participation in the review process.

Authors

---

### Author Response · Authors · 2025-12-02
**Conclusion of Discussion**

The preceding discussion was abruptly interrupted and resulted in an AC change due to information leakage. To facilitate the new AC in quickly reviewing the discussion phase, we provide the following summary.

All reviewers maintain a positive stance on our paper. Specifically, they all concur that our paper is well-written and clearly structured. Our experiments are not only comprehensive, covering diverse benchmarks, but also feature interesting visualizations and insightful conclusions. They believe that the open-sourcing of our model and dataset will serve as valuable assets to the research community.

We have addressed every Weakness and all Questions raised by the reviewers with corresponding explanations or supplementary experiments. All concerns can be broadly categorized into the following three areas:

1.  **Detailed Analysis of the Point Cloud Modality.** Reviewers observed that our representation model's performance on the point cloud (PC) modality was comparatively weaker than other modalities in real-world robotics tasks. They proposed three directions for investigation: (a) whether other pre-trained PC representation models suffer from the same issue, (b) whether adopting a more advanced PC encoder architecture could resolve it, and (c) whether PC quality enhancement could be a solution. Our supplementary experiments revealed that other pre-trained point cloud models indeed face this problem, and a more advanced encoder structure did not provide a fix. Crucially, we found that point cloud quality enhancement significantly improved our model's performance, while having a negligible impact on the baseline. The participating reviewers were satisfied with these additional experiments.
2.  **Comparison with VGGT.** Given that VGGT (CVPR 2025 Best Paper) demonstrates extremely strong 3D estimation capabilities, the reviewers requested a comparison of its performance as a representation model for robot manipulation tasks. Our supplementary experiments demonstrated that VGGT, when used as a representation model, does not yield significant improvements in policy performance and performs worse than EmbodiedMAE. We had a multi-round discussion with Reviewer RVAu on this matter, who expressed satisfaction and confirmed they would raise their score.
3.  **Extending Experiments.** This included expanding experiments related to the ACT Policy, performing ablation studies on dataset scale, etc. We have completed all the requested experiments and incorporated them into the revision.

In summary, the two reviewers who participated in the discussion both expressed satisfaction with the rebuttal and indicated a score increase. We are confident that our responses will also well address the concerns of the two reviewers who did not participate in the discussion. We believe all of our reviewers are **responsible and professional**. We had a pleasant discussion, and all score increases occurred **prior to the information leakage event**, with **no inappropriate behavior**.

---

### Meta-Review · Area_Chair_hiph · 2026-01-10

**Summary:**

The paper presents EmbodiedMAE, a 3D multi-modal representation model for robot manipulation, alongside the DROID-3D dataset. Reviewers appreciated the clear writing, comprehensive experiments, and public dataset release. Strengths include a unified multi-modal approach, robust evaluation across simulation and real robots, and a scalable distillation strategy. However, key weaknesses remain: the point cloud modality underperforms without strong justification, the necessity of MAE-based pretraining over alternatives is not convincingly argued, inference latency and practical deployment considerations are insufficiently addressed, and certain ablations (e.g., Dirichlet parameter, PC encoder variants, single-modality contributions) are incomplete or missing.

**Reviewer Concerns:**

The rebuttal addressed several experimental concerns: enhanced point cloud preprocessing improved EmbodiedMAE-PC performance, sparse convolution alternatives were tested, subset training of DROID-3D showed minimal impact, and VGGT comparisons were clarified. Outstanding issues include the lack of ablations for MAE vs. other pretraining paradigms, incomplete analysis of the Dirichlet masking parameter, insufficient justification for tri-modal pretraining necessity, and limited discussion of latency and real-world feasibility. Additionally, the underperformance of point cloud-only policies remains only partially explained, leaving questions about robustness and general applicability.

**Reviewer Scores:**

If reviewers could fully participate in discussion after rebuttal, Reviewer 1 and 2 likely maintain a score at the marginally below acceptance threshold (4) due to unresolved core concerns. Reviewer 3, noting practical deployment and tri-modal justification issues, might revise the scores downward from marginally above threshold (6) to near threshold (5), reflecting residual methodological gaps. Overall, 3/4 reviewers acknowledged technical effort and clarity but shared doubts on robustness, novelty justification, and completeness of ablation studies, supporting a recommendation to reject.

---

### Decision · Program_Chairs · 2026-01-26

Reject